# Immunopeptidomics reveals determinants of *Mycobacterium tuberculosis* antigen presentation on MHC class I

Owen Leddy[1,2,3], Forest M White[1,3,4†], Bryan D Bryson[1,2*†]

[1]Department of Biological Engineering, Massachusetts Institute of Technology, Cambridge, United States; [2]Ragon Institute of Massachusetts General Hospital, Harvard, and MIT, Cambridge, United States; [3]Koch Institute for Integrative Cancer Research, Cambridge, United States; [4]Center for Precision Cancer Medicine, Cambridge, United States

**Abstract** CD8+ T cell recognition of *Mycobacterium tuberculosis* (*Mtb*)-specific peptides presented on major histocompatibility complex class I (MHC-I) contributes to immunity to tuberculosis (TB), but the principles that govern presentation of *Mtb* antigens on MHC-I are incompletely understood. In this study, mass spectrometry (MS) analysis of the MHC-I repertoire of *Mtb*-infected primary human macrophages reveals that substrates of *Mtb*'s type VII secretion systems (T7SS) are overrepresented among *Mtb*-derived peptides presented on MHC-I. Quantitative, targeted MS shows that ESX-1 activity is required for presentation of *Mtb* peptides derived from both ESX-1 substrates and ESX-5 substrates on MHC-I, consistent with a model in which proteins secreted by multiple T7SSs access a cytosolic antigen processing pathway via ESX-1-mediated phagosome permeabilization. Chemical inhibition of proteasome activity, lysosomal acidification, or cysteine cathepsin activity did not block presentation of *Mtb* antigens on MHC-I, suggesting involvement of other proteolytic pathways or redundancy among multiple pathways. Our study identifies *Mtb* antigens presented on MHC-I that could serve as targets for TB vaccines, and reveals how the activity of multiple T7SSs interacts to contribute to presentation of *Mtb* antigens on MHC-I.

## Editor's evaluation

This landmark study uses compelling approaches such as quantitative and screening mass spectrometry to identify peptides from tuberculosis bacteria that are presented by macrophages infected with this pathogen. The authors provide convincing evidence that the presentation of these antigens depends on a specialist bacterial secretion system. The study will be of interest to infectious disease specialists and of particular value for future vaccine development.

## Introduction

Tuberculosis (TB), caused by *Mycobacterium tuberculosis* (*Mtb*), is a leading cause of infectious disease mortality worldwide, causing approximately 10 million new cases of active TB disease and 1.5 million deaths per year (*World Health Organization 2022, 2021*). Currently, the only clinically licensed vaccine to prevent TB is Bacille Calmette-Guerin (BCG), which protects children against disseminated *Mtb* infection (*Roy et al., 2014*), but provides limited and highly variable protection against pulmonary TB in adults (*Fine, 1995*). More effective vaccines against TB are therefore needed, but identifying *Mtb* antigens capable of eliciting protective immunity remains challenging.

**\*For correspondence:**
bryand@mit.edu

†These authors contributed equally to this work

**Competing interest:** The authors declare that no competing interests exist.

Multiple convergent lines of evidence from experiments in mouse and non-human primate models of TB show that CD8+ T cells can contribute to immune control of *Mtb* infection (*Chen et al., 2009*; *Flynn et al., 1992*; *Woodworth et al., 2008b*), but the antigenic targets of protective CD8+ T cell immunity to *Mtb* infection have not been conclusively defined. In murine models, CD8+ T cells specific for some immunodominant *Mtb* antigens poorly recognize *Mtb*-infected macrophages (*Yang et al., 2018*), implying that infected macrophages may not present all *Mtb* antigens that elicit cytokine-producing CD8+ T cell responses (*Kamath et al., 2004*; *Lewinsohn et al., 2017*). These results suggest a need to directly identify which *Mtb* antigens are presented on MHC-I by infected phagocytes (*Flynn et al., 2011*).

It is currently unknown which *Mtb* proteins are able to enter MHC-I antigen processing pathways in macrophages infected with virulent *Mtb*. Whereas some bacterial species are lysed following phagocytosis and expose their internal cell contents to antigen processing pathways (*Shen et al., 1998*), a high proportion of virulent *Mtb* remains intact and viable in macrophages (*Armstrong and Hart, 1975*; *Lee et al., 2008*; *Lewis et al., 2003*), leading us to hypothesize that only a subset of *Mtb* proteins may be accessible for processing and presentation on MHC-I. Here, we use MS-based identification of peptides bound to MHC-I (immunopeptidomics) to directly identify *Mtb*-derived peptides presented on MHC-I in primary human macrophages infected with virulent *Mtb* H37Rv, revealing potential targets for CD8+ T cell-mediated immunity. Additionally, we use targeted MS to quantify changes in the presentation of *Mtb* peptides resulting from genetic perturbations to *Mtb* and chemical perturbations to the host cell, allowing us to probe host and bacterial determinants of antigen presentation on MHC-I in *Mtb* infection.

## Results

To identify *Mtb* antigens presented on MHC-I, we infected primary human monocyte-derived macrophages with *Mtb* H37Rv, isolated MHC-I by immunoprecipitation 72 hours post-infection, purified the associated peptides, and identified peptides by liquid chromatography coupled to tandem mass spectrometry (LC-MS/MS). We selected the 72 hour time point over shorter time points to allow sufficient time for antigen processing and presentation on MHC-I. Past experience suggested that choosing a longer time point would have resulted in a high rate of cell death. For an initial set of three analyses, we adapted a previously described protocol (*Stopfer et al., 2020*; Protocol 1 – see Methods) for use in a biosafety level 3 (BL3) setting. We developed a further optimized protocol (Protocol 2 – see Methods) that we used for 3 additional analyses (*Figure 1A*). The class I human leukocyte antigen (HLA) genotypes of the 6 donors analyzed were largely distinct from one another (*Table 1*), enabling us to identify peptides associated with a variety of HLA alleles.

We identified thousands of peptides for each primary cell donor, with a length distribution typical of MHC-I peptides (*Figure 1—figure supplement 1A*) and retention times that correlated well with hydrophobicity (*Figure 1—figure supplement 1B*). Unsupervised clustering of identified peptides using GibbsCluster 2.0 (*Andreatta et al., 2017*) revealed groups corresponding to the known peptide sequence binding motifs of class I HLA alleles expressed by each donor (*Figure 1—figure supplement 2*), confirming the specificity of our pulldowns. *Mtb*-derived MHC-I peptides detected in each pulldown were predicted to bind at least one class I HLA allele expressed by the donor (*Figure 1—figure supplement 3*). *Mtb* peptides made up less than 0.1% of MHC-I peptides identified, and this proportion was not discernibly increased by pre-treating macrophages with IFN-γ or treating with cycloheximide to inhibit host protein synthesis (*Figure 1B*).

Putative *Mtb* peptides that passed manual inspection of MS/MS spectra and extracted ion chromatograms (see Methods) were further validated using internal standard parallel reaction monitoring (IS-PRM, also known as SureQuant) (*Gallien et al., 2015*; *Stopfer et al., 2021a*; *Figure 1—figure supplement 4*). Putative *Mtb* peptides that had MS/MS spectra that closely matched that of a synthetic stable isotope-labeled (SIL) standard, co-eluted with the SIL standard, and were not detected in mock-infected control samples by SureQuant were considered correctly identified, authentic *Mtb* peptides (*Figure 1—figure supplement 5*). 77.85% of MASCOT identifications of putative *Mtb* peptides were rejected after manual data inspection, and of the remaining candidates a further 51.51% were rejected following analysis by SureQuant (*Table 2*), highlighting the need for rigorous validation using best practices (*Fritsche et al., 2021*) when identifying pathogen-derived peptides among an MHC repertoire dominated by host-derived peptides.

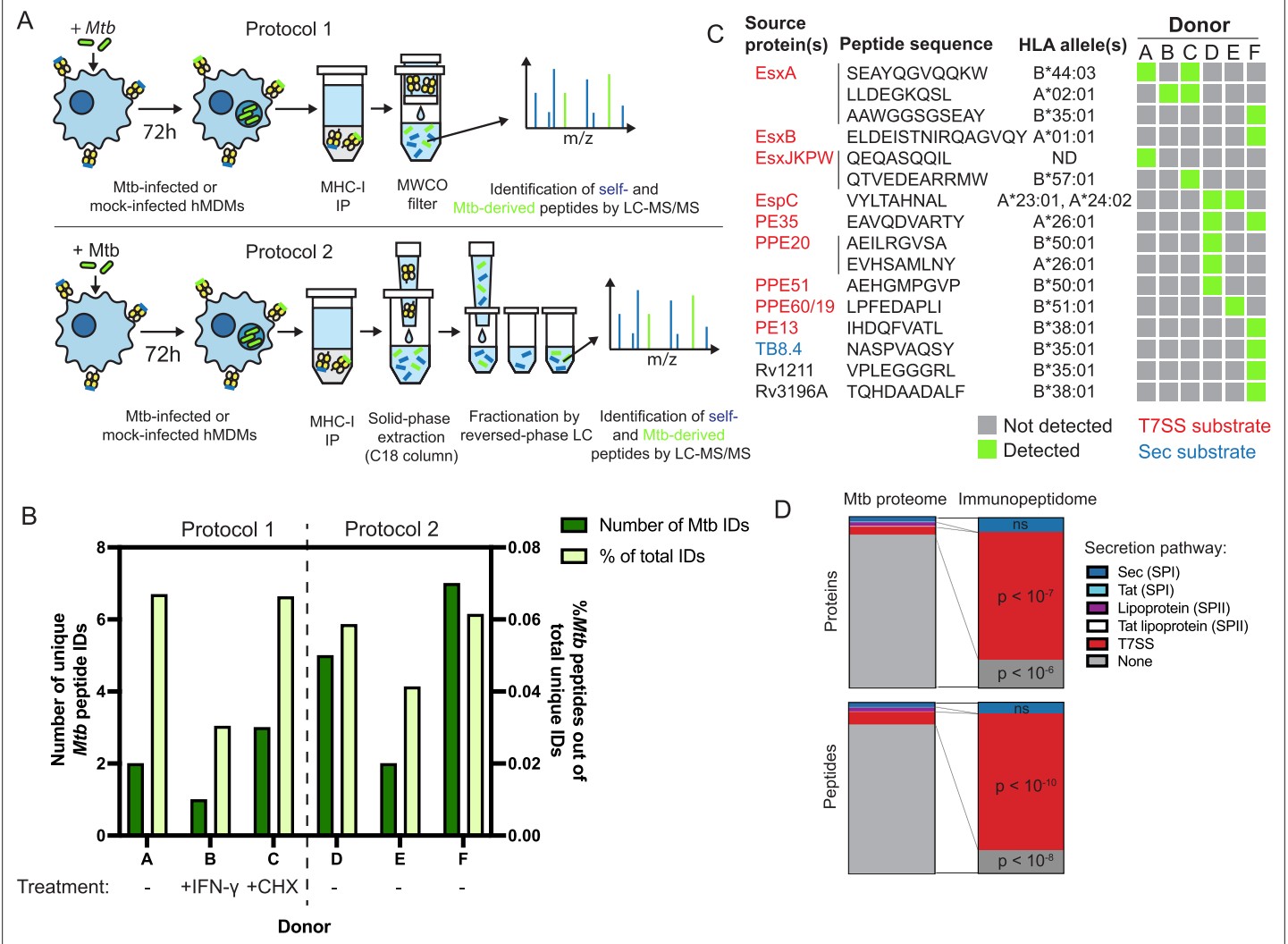

**Figure 1.** The MHC-I immunopeptidome of *Mtb*-infected human macrophages is enriched for T7SS substrates. (**A**) Schematic representation of two immunopeptidomics workflows used to profile the MHC-I repertoire of *Mtb*-infected primary human macrophages. IP: immunoprecipitation. MWCO: molecular weight cutoff. LC-MS/MS: liquid chromatography coupled to tandem mass spectrometry. (**B**) Absolute and relative number of *Mtb*-derived MHC-I peptides identified for each donor. Macrophages from donor B were pre-treated for 24 hr with 10 ng/mL IFN-γ, and macrophages from donor C were treated with 0.5 μg/mL cycloheximide (CHX) for the final 6 hr of infection. (**C**) Sequences, source proteins, associated HLA alleles, and donors for each validated *Mtb*-derived MHC-I peptide. (**D**) Enrichment analysis of *Mtb* peptides presented on MHC-I and their source proteins, categorized by protein secretion pathway using SignalP 6.0 (*Teufel et al., 2022*) and a curated set of known or strongly suspected T7SS substrates. p-Values for enrichment analyses of proteins and peptides were determined using the binomial test and the hypergeometric test respectively (see Methods).

The online version of this article includes the following source data and figure supplement(s) for figure 1:

**Source data 1.** Source data for enrichment analysis.

**Figure supplement 1.** Performance of two BSL-3 compatible MHC-I immunopeptidomics workflows.

**Figure supplement 2.** Gibbs clustering groups MHC-I peptides into clusters that correspond to HLA alleles expressed by each donor.

**Figure supplement 3.** Predicted class I HLA binding of *Mtb*-derived MHC-I peptides.

**Figure supplement 4.** Workflow for validating *Mtb*-derived MHC-I peptide identifications by SureQuant.

**Figure supplement 5.** Validation of *Mtb*-derived MHC-I peptide identifications (VYLTAHNAL, EAVQDVARTY, EVHSAMLNY, AEILRGVSA).

**Figure supplement 6.** Validation of *Mtb*-derived MHC-I peptide identifications (AEHGMPGVP, LPFEDAPLI, AAWGGSGSEAY, ELDEISTNIRQAGVQY).

**Figure supplement 7.** Validation of *Mtb*-derived MHC-I peptide identifications (IDHQFVATL, NASPVAQSY, VPLEGGGRL, TQHDAADALF).

**Figure supplement 8.** Validation of *Mtb*-derived MHC-I peptide identifications (SEAYQGVQQKW, QEQASQQIL, LLDEGKQSL, QTVEDEARRMW).

**Table 1.** Class I HLA alleles expressed by primary monocyte donors.

| Donor | HLA-A allele 1 | HLA-A allele 2 | HLA-B allele 1 | HLA-B allele 2 | HLA-C allele 1 | HLA-C allele 2 |
|-------|----------------|----------------|----------------|----------------|----------------|----------------|
| A | ND | ND | ND | ND | ND | ND |
| B | 02:01 | 24:02 | 07:02 | 27:07 | 07:02 | 15:02 |
| C | 01:01 | 02:01 | 44:03 | 57:01 | 06:02 | 16:01 |
| D | 23:01 | 26:01 | 35:01 | 50:01 | 04:01 | 06:02 |
| E | 11:01 | 24:02 | 39:06 | 51:01 | 07:02 | 15:02 |
| F | 01:01 | 26:01 | 35:01 | 38:01 | 04:01 | 12:03 |

Of the 16 *Mtb*-derived MHC-I peptides we identified, 13 (81.25%) derived from proteins secreted via type VII secretion systems (T7SS) (*Figure 1C*). The *Mtb* genome encodes five of these protein export machines (designated ESX-1, 2, 3, 4, and 5; *Abdallah et al., 2007*), and we identified MHC-I peptides derived from proteins known to be secreted by three of these systems [ESX-1 (*Guinn et al., 2004*; *Millington et al., 2011*; *Stanley et al., 2003*), ESX-3 (*Siegrist et al., 2009*; *Tufariello et al., 2016*), and ESX-5 (*Abdallah et al., 2006*; *Daleke et al., 2012*; *Ekiert and Cox, 2014*; *Shah et al., 2015*)]. For several T7SS substrates, we identified multiple peptides from the same protein, and/or identified the same peptide across multiple donors. These antigens included EsxA, PPE20, EspC, and PE35, as well as sequences conserved among the four nearly identical members of the EsxJ family of proteins (EsxJ, EsxK, EsxP, and EsxW – referred to here as EsxJKPW). T7SS substrates were significantly overrepresented in the MHC-I repertoire relative to the whole *Mtb* proteome, both at the peptide level ($p<10^{-10}$; binomial test with Bonferroni correction) and at the protein level ($p<10^{-7}$; hypergeometric test with Bonferroni correction; *Figure 1D*). Proteins without identifiable secretion signals were significantly underrepresented among *Mtb*-derived MHC-I peptides ($p<10^{-8}$; binomial test with Bonferroni correction) and source proteins ($p<10^{-6}$; hypergeometric test with Bonferroni correction). While some of the *Mtb* antigens we identified are highly abundant compared to the rest of the *Mtb*

**Table 2.** Putative *Mtb*-derived peptides that passed manual inspection of DDA MS data but failed SureQuant validation.

| Source protein | Peptide sequence |
|----------------|------------------|
| Rv0383c | AAPGRPVAPG |
| pyrD | GDRLALISV |
| Rv2303c | KHPNVYLEL |
| Rv0839 | YTHGYHES |
| kgd | AERAAAAAP |
| Rv1375 | EAAQSRITA |
| GabD1 | AKVGASAAY |
| PE1 | AAGNLRAAI |
| HlfX | IPYDRGDLV |
| Rv2807 | AKWILEGIK |
| Rv3818 | IAPELVRT |
| Rv1065 | YTRIHGDEEL |
| Rv0293c | DELIAGLAY |
| Rv3779 | VAIAVGPALT |
| PPE55 | TVAPINLNP |
| Rv2263 | QEIEEGIL |
| Rv0333 | GEDPGIAR |

proteome (e.g. EsxB), the abundances of other T7SS substrates are near the mean (*Figure 1—figure supplement 1C*; *Schubert et al., 2015*), suggesting that the overrepresentation of T7SS substrates is not solely due to greater abundance relative to the rest of the *Mtb* proteome. These results suggest that T7SS substrates may preferentially gain access to MHC-I antigen processing pathways.

To nominate possible mechanisms by which *Mtb* antigens might be processed and loaded onto MHC-I, we used confocal microscopy to examine the intracellular fate of *Mtb* in primary human macrophages. We hypothesized that *Mtb* antigens could either (1) be processed by endolysosomal proteases and loaded onto MHC-I in *Mtb*-containing compartments, or (2) gain access to cytosolic antigen processing pathways via permeabilization of the phagosome membrane (*Grotzke et al., 2010*; *Grotzke et al., 2009*; *van der Wel et al., 2007*; *Figure 2A*). In a subset of macrophages at both early and late timepoints, a subset of *Mtb* co-localized with Galectin-3, a marker of phagosomal membrane permeabilization (*Chauhan et al., 2016*; *Watson et al., 2012*; *Figure 2B–C*; *Figure 2—figure supplement 1A*). A subset of *Mtb* also co-localized with P62, an autophagy adaptor protein that recognizes cytosol-exposed bacteria (*Watson et al., 2012*; *Zheng et al., 2009*; *Figure 2B–C*; *Figure 2—figure supplement 1B*). Since bacteria become ubiquitinated upon exposure to the cytosol and recruitment of autophagy adaptor proteins to the *Mtb* phagosome has previously been associated with damage to the phagosomal membrane (*Watson et al., 2012*), these results suggest that *Mtb* gains access to the host cytosol in primary human macrophages, as has previously been shown in murine macrophages (*Mittal et al., 2018*) and in cell lines (*Beckwith et al., 2020*). A subset of *Mtb*-containing phagosomes co-localized with the late endosome and lysosome marker LAMP-1 (*Figure 2B–C*; *Figure 2—figure supplement 1C*), suggesting access to endolysosomal proteases, but *Mtb*-containing phagosomes did not co-localize with MHC-I itself (*Figure 2B–C*; *Figure 2—figure supplement 1D*), suggesting that *Mtb* antigens were unlikely to be loaded onto MHC-I in the *Mtb*-containing phagosome. These results suggested that access to the host cell cytosol represented a likely route for processing and presentation of *Mtb* antigens. An *Mtb* strain deficient in the activity of the ESX-1 secretion system (*eccCa1:Tn*) exhibited reduced co-localization with Galectin-3 and P62 (*Figure 2D–F*). Our results are consistent with prior studies demonstrating that ESX-1 activity is required for *Mtb* to damage the phagosome membrane (*Augenstreich et al., 2017*; *Simeone et al., 2012*; *Watson et al., 2012*) and show that this phenomenon occurs in our primary human macrophage infection model.

Our microscopy results led us to hypothesize that ESX-1-mediated phagosomal membrane damage might be required for *Mtb* antigens to access MHC-I antigen processing pathways (*Figure 3A*). If this were the case, ESX-1 activity would be required for presentation not only of peptides derived from ESX-1 substrates (e.g. EsxA$_{28-36}$ – sequence LLDEGKQSL), but also peptides derived from substrates of other T7SSs (e.g. EsxJKPW$_{24-34}$ – sequence QTVEDEARRMW – which is derived from ESX-5 substrates that do not require ESX-1 for secretion; *Champion et al., 2006*). To test this hypothesis, we turned to quantitative targeted MS (SureQuant) to quantify changes in the presentation of EsxA$_{28-36}$ and EsxJKPW$_{24-34}$ across multiple experimental conditions (*Figure 3B*). We used primary macrophages from donors expressing HLA-A*02:01 and HLA-B*57:01 for these experiments to ensure presentation of the target peptides. Because the other *Mtb* epitopes detected in our untargeted MS experiments are not expected to bind these HLA alleles, we only targeted EsxA$_{28-36}$ and EsxJKPW$_{24-34}$. While this targeted approach is limited in the number of epitopes it can detect, it enables reliable and accurate quantification of peptides across experimental conditions with low sample input. Because methionine residues of peptides can oxidize during sample handling, we targeted both the oxidized and non-oxidized form of EsxJKPW$_{24-34}$ where possible. In addition to the SIL synthetic trigger peptides required for SureQuant, we also spiked pre-formed heavy isotope labeled peptide-MHC complexes (hipMHCs) into the lysates prior to immunoprecipitation to provide an internal standard that can be used to normalize out technical variation, which improves the accuracy of label-free quantification (*Stopfer et al., 2020*).

Targeted MS analysis revealed that macrophages infected with ESX-1-deficient *Mtb* (*eccCa1:Tn*) did not present either EsxA$_{28-36}$ or EsxJKPW$_{24-34}$ on MHC-I (*Figure 3c*). We assessed alternative explanations for this difference in presentation of EsxA$_{28-36}$ and EsxJKPW$_{24-34}$ and observed no difference in *Mtb* outgrowth (*Figure 3D*), the proportion of cells infected (*Figure 3—figure supplement 1A–B*), or total MHC-I surface expression (*Figure 3—figure supplement 1C–D*) between macrophages infected with wild-type or ESX-1-deficient *Mtb*. These results show that ESX-1 activity is required for presentation of EsxA$_{28-36}$ and EsxJKPW$_{24-34}$ on MHC-I in macrophages.

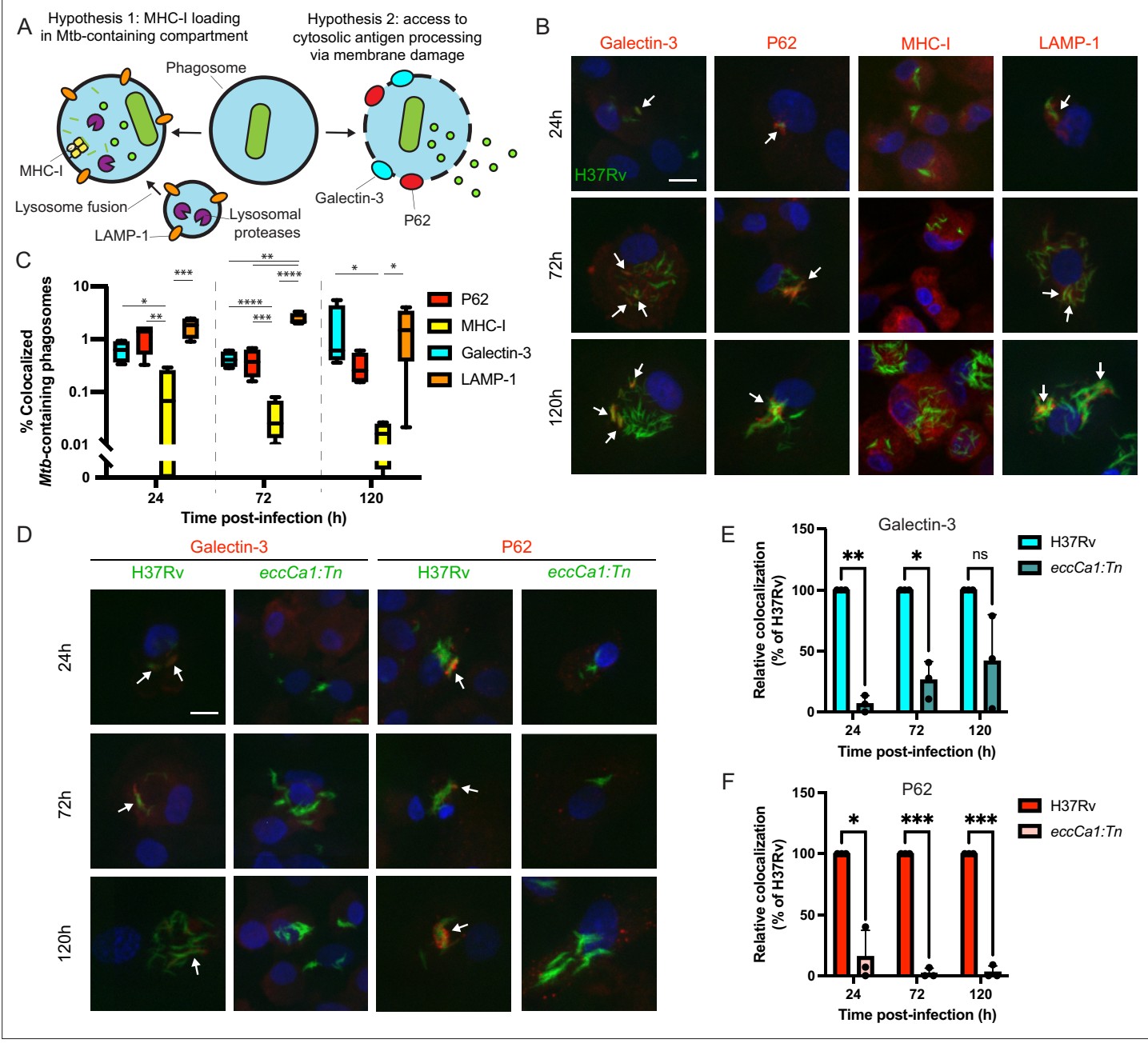

**Figure 2.** *Mtb* co-localizes with markers of phagosome membrane damage in an ESX-1-dependent manner and does not co-localize with MHC-I. Primary human macrophages were infected with GFP-expressing wild-type *Mtb*, fixed at 24, 72, or 120 hr post-infection, stained by immunofluorescence (IF), and imaged by spinning-disk confocal microscopy. (**A**) Schematic showing markers associated with each of two possible pathways of *Mtb* antigen processing and presentation. (**B**) Representative images of *Mtb*-infected macrophages stained for Galectin-3, P62, MHC-I, or LAMP-1. Scale bar indicates 10 µm. White arrows indicate *Mtb*-containing phagosomes co-localizing with each marker. (**C**) Automated quantification (see Methods) of the proportion of co-localized *Mtb*-containing phagosomes for each marker for n=4 donors (* p<0.05, ** p<0.01, *** p<0.001, **** p<0.0001; one-way ANOVA with Tukey's multiple comparisons test). (**D**) Representative images of macrophages infected with wild-type (H37Rv) or ESX-1-deficient (*eccCa1:Tn*) *Mtb* stained for Galectin-3 or P62. Scale bar indicates 10 µm. White arrows indicate *Mtb*-containing phagosomes co-localizing with each marker. (**E–F**) Automated quantification of the relative proportion of GFP +objects co-localized with IF staining for Galectin-3 (**E**) or P62 (**F**) as a function of time post-infection for n=3 donors, normalized to wild-type (H37Rv) (* p<0.05, ** p<0.01, *** p<0.001; paired t-test). Error bars indicate standard deviation.

The online version of this article includes the following source data and figure supplement(s) for figure 2:

**Source data 1.** Single-channel images.

*Figure 2 continued on next page*

*Figure 2 continued*

**Source data 2.** Colocalization quantification source data.

**Figure supplement 1.** Immunofluorescence staining for Galectin-3, P62, LAMP-1, and MHC-I is specific.

To determine whether the absence of EsxA$_{28-36}$ and EsxJKPW$_{24-34}$ in the MHC-I repertoire of macrophages infected with ESX-1-deficient *Mtb* could be attributed to a loss of type I interferon signaling (**Stanley et al., 2007**), we added exogenous IFN-β to macrophages infected with ESX-1-deficient *Mtb* and again quantified presentation of EsxA$_{28-36}$ and EsxJKPW$_{24-34}$ by SureQuant. Addition of exogenous IFN-β restored a type I interferon response as measured by production of CXCL10 (**Figure 3E**), but did not rescue presentation of EsxA$_{28-36}$ or EsxJKPW$_{24-34}$ on MHC-I (**Figure 3F**). These results show that presentation of EsxA$_{28-36}$ or EsxJKPW$_{24-34}$ on MHC-I is dependent on ESX-1 activity but independent of downstream type I interferon signaling, consistent with a model in which ESX-1-mediated phagosomal damage enables *Mtb* antigens to access MHC-I antigen processing pathways.

Many MHC-I peptides are proteolytically processed by the proteasome (**Kloetzel, 2001**), while others are processed by endosomal or lysosomal proteases (**Grotzke et al., 2017**; **Shen et al., 2004**). To determine whether these mechanisms contribute to presentation of *Mtb* peptides on MHC-I, we treated HLA-A*02:01+, HLA-B*57:01+ macrophages with inhibitors of proteasome activity (MG-132), cysteine cathepsin activity (E64d), and lysosomal acidification (bafilomycin; **Figure 4A**). We quantified presentation of EsxA$_{28-36}$ and EsxJKPW$_{24-34}$ on MHC-I along with five host-derived HLA-A*02:01-binding peptides identified in previous studies (**Stopfer et al., 2021b**) that could be reliably detected in the MHC-I repertoire of macrophages. We began drug treatment of the macrophages prior to infection with *Mtb* and limited the duration of infection to 24 hr so that the cells could be treated with drug for the full duration of the infection without excessive cytotoxicity (**Figure 4B**). Treatment with MG-132 inhibited proteasome activity, as measured by accumulation of proteins modified with K48-linked polyubiquitin (**Figure 4—figure supplement 1A–B**). Treatment with E64d inhibited cathepsin B activity, as measured by a fluorometric assay (**Figure 4—figure supplement 1C**). Treatment with bafilomycin inhibited lysosomal acidification, as measured by lysotracker staining (**Figure 4—figure supplement 1D–E**). All three drugs exhibited minimal cytotoxicity in macrophages at the doses used in our immunopeptidomic experiments (**Figure 4—figure supplement 2**), did not inhibit phagocytosis of *Mtb* or bacterial outgrowth (**Figure 4—figure supplement 3**), and did not have a significant effect on surface MHC-I levels (**Figure 4—figure supplement 3**).

Treatment with MG-132 and E64d each respectively reduced presentation of a subset of self MHC-I peptides but did not reduce presentation of *Mtb* peptides (**Figure 4C**). Treatment with bafilomycin broadly increased presentation of target MHC-I peptides (**Figure 4C**). This increase was statistically significant for EsxA$_{28-36}$ (**Figure 4D**; $p<0.05$, one-way ANOVA with Dunnett's multiple comparisons test), though not for EsxJKPW$_{24-34}$ (**Figure 4E**). Cell surface levels of HLA-DR and HLA-DQ were not significantly affected by bafilomycin or E64d treatment (**Figure 4—figure supplement 4**), suggesting that E64d and bafilomycin did not substantially stall MHC-II antigen presentation and their effects of on MHC-I presentation could not be explained as indirect effects of modulating antigen entry into the MHC-II antigen processing pathway. Our results suggest that processing of *Mtb* peptides for presentation on MHC-I relies on antigen processing proteases other than the proteasome, cysteine cathepsins, or other acidification-dependent lysosomal proteases, and/or that multiple redundant pathways contribute to processing of *Mtb* peptides for presentation on MHC-I.

## Discussion

Our analysis of the MHC-I repertoire of primary human macrophages infected with *Mtb* revealed that T7SS substrates are a prominent source of *Mtb* peptides presented on MHC-I. Our findings contrast with those of previous studies of the immunopeptidome of macrophage cell lines infected with BCG (**Bettencourt et al., 2020**) and the avirulent strain *Mtb* H37Ra (**Flyer et al., 2002**), in which the mycobacterial peptides identified included twin arginine translocation (Tat) pathway substrates, Sec pathway secretion substrates, membrane-associated proteins, and cytosolic proteins, with no apparent enrichment for T7SS substrates. Given that we showed ESX-1 activity contributes to presentation of *Mtb* antigens in infection with virulent *Mtb* H37Rv, the fact that BCG and *Mtb* H37Ra are both deficient in ESX-1 activity (**Frigui et al., 2008**; **Gordon et al., 1999**; **Guinn et al., 2004**; **Lewis**

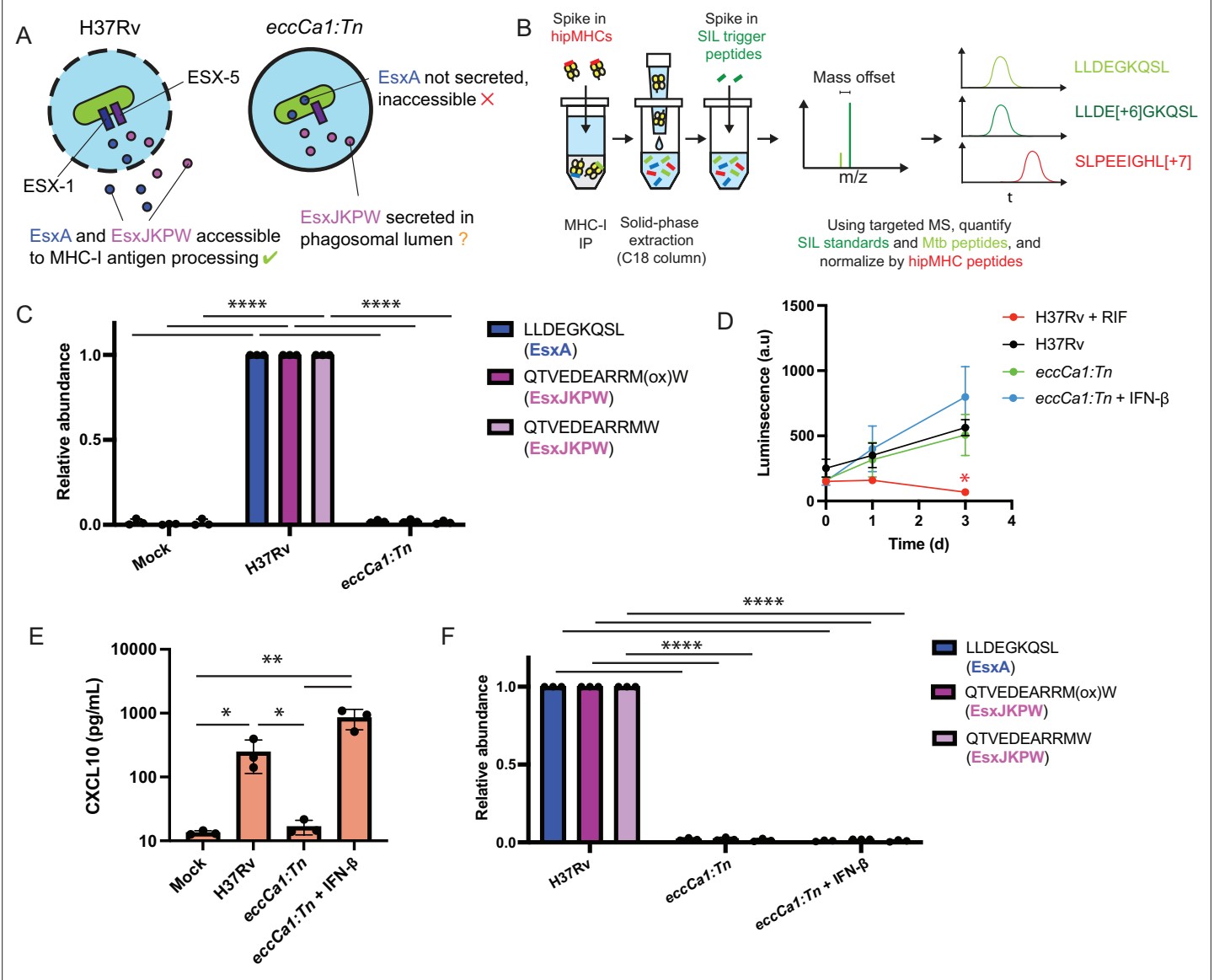

**Figure 3.** ESX-1 activity is required for presentation of EsxA$_{28-36}$ and EsxJKPW$_{24-34}$ on MHC-I, independently of type I interferon signaling. (**A**) Schematic representation of the localization of EsxA and EsxJKPW in macrophages infected with wild-type *Mtb* H37Rv or the ESX-1-deficient *eccCa1:Tn* transposon mutant. (**B**) Schematic showing our workflow for targeted detection and quantification of *Mtb*-derived MHC-I peptides by SureQuant, using stable isotope labeled peptide-MHC complexes (hipMHCs) as internal standards. SIL: stable isotope labeled. (**C**) Relative quantification of EsxA$_{28-36}$ and EsxJKPW$_{24-34}$ by SureQuant in macrophages infected with no *Mtb* (mock), wild-type *Mtb* H37Rv, or *eccCa1:Tn* for n=3 donors (all HLA-A*02:01+, HLA-B*57:01+). As oxidation of methionine is common during sample handling, both the oxidized and non-oxidized form of EsxJKPW$_{24-34}$ were quantified. (**D**) Luminescence as a function of time measured for macrophages infected with luciferase-expressing *Mtb*, in a wild-type H37Rv or *eccCa1:Tn* background, with or without the addition of 10 ng/mL IFN-β in the culture media. Addition of 25 µg/mL rifampicin (RIF) to the culture media was used as a control showing reduced luminescence with bacterial death. Data points and error bars represent the mean and standard deviation of n=3 donors, each of which represents the mean of three technical replicates. (* p<0.05, one-way ANOVA with Dunnett's multiple comparisons test, relative to H37Rv as the reference condition). (**E**) CXCL10 concentration in the culture media 72 hr post-infection quantified by ELISA. Data points each represent the mean of three technical replicates for a given donor. (* p<0.05, ** p<0.01, one-way ANOVA with Tukey's multiple comparisons test on log-transformed concentrations.) (**f**) Relative quantification of EsxA$_{28-36}$ and EsxJKPW$_{24-34}$ by SureQuant in macrophages infected with no *Mtb* (mock), wild-type *Mtb* H37Rv, or *eccCa1:Tn* for n=3 donors (all HLA-A*02:01+, HLA-B*57:01+). (**** p<0.001, one-way ANOVA with Tukey's multiple comparisons test.). Error bars indicate standard deviation.

The online version of this article includes the following source data and figure supplement(s) for figure 3:

**Source data 1.** Relative abundances of *Mtb*-derived MHC-I peptides determined by SureQuant.

**Source data 2.** CXCL10 ELISA raw data.

*Figure 3 continued on next page*

Figure 3 continued

**Figure supplement 1.** Infection of macrophages with wild-type or ESX-1-deficient *Mtb* strains results in similar rates of infection and does not affect surface MHC-I levels 72 hr post-infection.

**Figure supplement 1—source data 1.** Source data for plots.

*et al., 2003*; *Pym et al., 2002*) may in part explain this difference in the mycobacterial peptides presented on MHC-I. The absence of ESX-1 in BCG therefore could limit the ability of BCG to prime effective T cell responses against *Mtb*, not only because of the absence of antigens encoded in the ESX-1 locus but also because of altered or reduced presentation of other antigens (*Pym et al., 2003*).

Our findings contrast with previous work arguing that ESX-1 activity is dispensable for presentation of *Mtb* antigens on MHC-I (*Lewinsohn et al., 2006*; *Woodworth et al., 2008a*). ESX-1 activity is not required for *Mtb*-infected human monocyte-derived dendritic cells to present an epitope derived from TB8.4 (a Sec pathway substrate) to cognate T cells (*Lewinsohn et al., 2006*). While ESX-1 activity is required for priming of CD8+ T cell responses specific for EsxB in vivo in mice, it is dispensable for priming of CD8+ T cell responses specific for antigens exported via other secretion systems (such as TB8.4 and EsxH) (*Woodworth et al., 2008b*). While these prior results suggest that the requirements for presentation of *Mtb* antigens on MHC-I may vary among *Mtb* antigens and/or vary among antigen presenting cell types, our results show that ESX-1 activity is essential for the presentation of certain *Mtb* antigens in infected human macrophages, beyond ESX-1 substrates alone. Whereas in axenic culture, secretion of ESX-5 substrates is independent of ESX-1 function (*Champion et al., 2006*; *Shah and Briken, 2016*), the fact that presentation of peptides derived from an ESX-5 substrate on MHC-I requires ESX-1 activity supports the hypothesis that the localization of secreted *Mtb* proteins within a host cell may depend on the activity of multiple T7SSs. This dependence on multiple T7SSs has previously been shown for other ESX-5 substrates such as CpnT (*Izquierdo Lafuente et al., 2021*), and a functional interdependence between ESX-1 and ESX-3 (which respectively damage the phagosome membrane and prevent host membrane repair) has previously been proposed (*Mittal et al., 2018*). Our results suggest that the interdependence of *Mtb* T7SSs in an intracellular context influences the availability of *Mtb* antigens for processing and presentation on MHC-I.

Prior studies have shown that *Mtb* antigens must access the cytosol to be presented on MHC-I in human monocyte-derived dendritic cells (*Grotzke et al., 2010*; *Grotzke et al., 2009*; *Lewinsohn et al., 2006*), and our results are consistent with a model in which ESX-1-mediated phagosomal membrane damage enables *Mtb* antigens to access cytosolic antigen processing and presentation pathways. However, further evidence will be needed to establish whether phagosomal membrane damage alone is sufficient to enable presentation of *Mtb* antigens on MHC-I in the absence of other ESX-1 functions. Given that our quantitative MS experiments only targeted two *Mtb*-derived epitopes, further experiments will also be needed to determine whether these results extend to all *Mtb* T7SS substrates or all *Mtb* MHC-I antigens, or only a subset.

In this study, we used strain H37Rv for the sake of consistency with other prior studies on antigen presentation on MHC-I and phagosome membrane damage in *Mtb*-infection, but antigen presentation may vary among *Mtb* isolates. *Mtb* isolates have been shown to differ in their level of ESX-1 activity (*Solans et al., 2014*), as well as carrying mutations in Esx-family proteins and other T7SS substrates (*Saelens et al., 2022*; *Uplekar et al., 2011*). Further studies across multiple isolates may therefore reveal differences in antigen presentation that could be relevant for the design of broadly protective vaccines.

We showed that presentation of two *Mtb*-derived peptides on MHC-I was independent of the activity of proteases commonly associated with antigen processing (the proteasome and cysteine cathepsins), suggesting an alternate processing mechanism. Other proteases have previously been proposed to contribute to proteolytic processing of antigens for presentation on MHCs, including tripeptidyl peptidase II (*Geier et al., 1999*; *Guil et al., 2006*; *Lázaro et al., 2015*; *York et al., 2006*), nardilysin (*Kessler et al., 2011*), thimet oligopeptidase (*Kessler et al., 2011*), metalloproteinases (*Lorente et al., 2012*), and serine cathepsins such as Cathepsin G that are not inhibited by E64d (*Burster et al., 2010*). Some of these or other enzymes may be responsible for proteolytic processing of EsxA$_{28-36}$ and EsxJKPW$_{24-34}$, or may have a redundant role that can compensate for inhibition of conventional antigen processing pathways. Further work will be needed to determine which proteolytic pathways contribute to presentation of *Mtb* T7SS substrates on MHC-I.

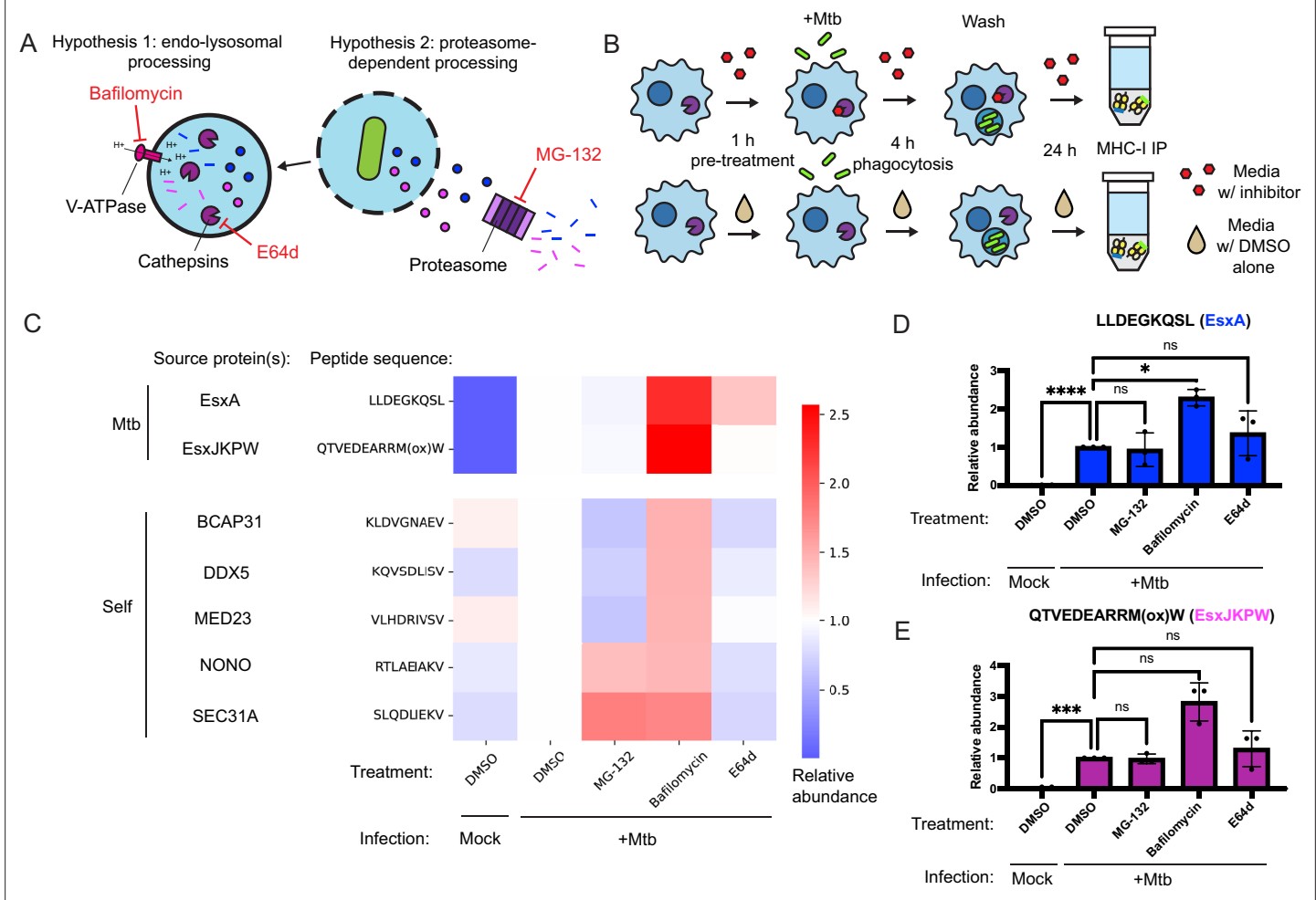

**Figure 4.** Inhibition of conventional antigen processing proteolytic pathways does not impair presentation of EsxA$_{28-36}$ and EsxJKPW$_{24-34}$ on MHC-I. (**A**) Schematic representation of proteolytic pathways inhibited by the proteasome inhibitor MG-132, the V-type ATPase inhibitor bafilomycin, and the cysteine cathepsin inhibitor E64d. (**B**) Schematic showing the timing of drug treatment and *Mtb* infection for targeted MS experiments. (**C**) Heatmap showing relative abundance of self and *Mtb*-derived MHC-I peptides determined by SureQuant in mock-infected macrophages or *Mtb*-infected macrophages treated with MG-132, bafilomycin, E64d, or DMSO-only control. Colors represent the mean fold change relative to the DMSO-treated, *Mtb*-infected condition for n=3 donors (all HLA-A*02:01+, HLA-B*57:01+). (**D–E**) Relative abundance of EsxA$_{28-36}$ (**D**) and EsxJKPW$_{24-34}$ (**E**) determined by SureQuant in mock-infected macrophages or *Mtb*-infected macrophages treated with MG-132, bafilomycin, E64d, or DMSO-only control for n=3 donors (all HLA-A*02:01+, HLA-B*57:01+). Error bars indicate standard deviation.

The online version of this article includes the following source data and figure supplement(s) for figure 4:

**Source data 1.** Relative abundances of MHC-I peptides determined by SureQuant.

**Figure supplement 1.** MG-132, E64d, and bafilomycin effectively inhibit their targets in *Mtb*-infected macrophages.

**Figure supplement 1—source data 1.** Uncropped western blot images.

**Figure supplement 1—source data 2.** Source data for plots.

**Figure supplement 2.** MG-132, E64d, and bafilomycin have minimal cytotoxicity at effective doses in *Mtb*-infected macrophages.

**Figure supplement 2—source data 1.** Source data for plots.

**Figure supplement 3.** Treatment of macrophages with MG-132, E64d, and bafilomycin does not impair phagocytosis or outgrowth of *Mtb*.

**Figure supplement 3—source data 1.** Source data for plots.

**Figure supplement 4.** Treatment of macrophages with E64d and bafilomycin does not significantly reduce surface levels of MHC-II.

**Figure supplement 4—source data 1.** Source data for plots.

In addition to T7SS substrates, we also identified peptides derived from a Sec pathway substrate (TB8.4), and from uncharacterized hypothetical proteins (Rv1211, Rv3196A). Rv1211 and Rv3196A have both previously been detected by MS in culture filtrates of *Mtb* (*Bell et al., 2012*), as has the *Mycobacterium marinum* ortholog of Rv3196A (MMAR_1367; *Cronin et al., 2022*), suggesting that either or both of these proteins could be secreted despite lacking readily identifiable secretion signals. TB8.4, Rv1211, and Rv3196A are all low-molecular-weight proteins, which we speculate could make it easier for them to translocate through pores in a permeabilized phagosomal membrane. Further investigation of the localization of these proteins may help us better understand how and to what extent small secreted *Mtb* proteins other than known T7SS substrates can contribute to the MHC-I repertoire.

Some of the *Mtb* MHC-I peptides we identified derive from proteins known or suspected to be associated with the mycobacterial outer membrane (*Abdallah et al., 2006*; *Sampson et al., 2001*). These include PPE51 (*Wang et al., 2020*), PPE20 (*Boradia et al., 2022*), PPE60 (*Su et al., 2018*), EspC (*Champion et al., 2009*; *Lou et al., 2017*), and PE13 [the co-transcribed binding partner (*Chen et al., 2017*) of the putative outer membrane protein PPE18 (*Nair et al., 2009*)]. While some of these proteins can also be found in soluble form in culture filtrates (*Tufariello et al., 2016*), others may be stably associated with the mycobacterial outer membrane (*Wang et al., 2020*). The mechanism by which antigens associated with the outer membrane are processed and presented could differ from that of soluble secreted proteins.

Prior work shows that at least some of the MHC-I antigens we identified as potential vaccine targets are immunogenic in humans. *Lewinsohn et al., 2017* showed that in humans with prior exposure to *Mtb*, EsxA, EsxB, PPE51, and EsxJKPW are sources of immunodominant CD8+ T cell antigens, while other antigens identified in our study elicited intermediate responses (PE13, PE35, PPE60, PPE19), and one elicited no detectable response (PPE20). In a separate study, CD8+ T cells (as well as CD4+ T cells) recognized multiple epitopes derived from EspC (*Millington et al., 2011*). Individual peptides we identified by MS have also been previously shown to be recognized by human T cells, including EsxJ$_{24-34}$ (*Grotzke et al., 2010*; *Lewinsohn et al., 2013*) and EsxA$_{28-36}$ (*Tully et al., 2005*), providing a proof of concept that particular epitopes identified by MS can be immunogenic. Antigens like PPE20 that did not elicit measurable CD8+ T cell responses in the context of natural infection should not necessarily be ruled out as vaccine targets, as a lack of CD8+ T cell recognition in the setting of natural infection does not necessarily imply that these antigens could not be immunogenic. Bioinformatic studies have also shown that predicted class I HLA-binding motifs are statistically underrepresented in several Esx-family protein sequences, suggesting that these proteins may be under selective pressure to escape recognition by CD8+ T cells (*Maman et al., 2011*). This finding provides further support for the idea that T cell recognition of T7SS substrates presented on MHC-I is functionally relevant to protective immunity against *Mtb* infection.

We did not detect any peptides derived from several antigens previously shown to stimulate CD8+ T cell responses in *Mtb* infection, including responses restricted to HLA alleles represented in our DDA MS analyses [for example, EsxH and FbpB (*Axelsson-Robertson et al., 2015*)]. This could be due to any of several biological and/or technical reasons. Given that the *Mtb*-derived MHC-I peptides we detected were enriched for T7SS substrates but not for substrates of the Sec or Tat secretion pathways, it is possible that the subcellular localization of antigens secreted via these pathways is not optimal for efficient presentation on MHC-I in macrophages. In this case, other types of APCs that differ in their antigen processing and presentation capabilities could still present peptides from these antigens to prime the previously observed CD8+ T cell responses. The same could also be true of certain T7SS substrate antigens (such as EsxH) if T7SS substrates vary in their subcellular localization or if presentation of certain epitopes requires antigen processing pathways not active in macrophages. In future studies, isolating MHC peptides from multiple myeloid cell types may help determine whether heterogeneity in antigen presentation among cell types contributes to apparent discrepancies between the *Mtb*-specific CD8+ T cell repertoire and the macrophage pMHC repertoire. Some peptides may also have been missed because they were lost to adsorption during sample handling, because they were below the limit of detection due to low abundance or ionization efficiency, or because they co-eluted with other, more abundant peptides and were therefore not prioritized for acquisition in DDA analyses. In future work, pMHC internal standards added prior to immunoprecipitation may enable accurate estimates of the limit of MS detection for specific peptides, to determine

whether technical factors led to a lack of detection. Our results do not rule out presentation of additional categories of antigens besides those we detected.

Our results identify several *Mtb* proteins that are accessible for presentation on MHC-I in *Mtb*-infected human macrophages, suggesting that these could serve as targets for subunit vaccines designed to elicit CD8+ T cell-mediated protection against TB. By showing that T7SS substrates are overrepresented in the MHC-I repertoire of macrophages and revealing mechanistic determinants of *Mtb* antigen processing, our results have the potential to guide future screens to identify additional vaccine targets. Our findings advance the field's understanding of antigen presentation in TB and suggest novel potential targets for TB vaccine development.

## Methods

### Human cell isolation, differentiation, and culture

Deidentified buffy coats were obtained from Massachusetts General Hospital. Samples are acquired and provided to research groups with no identifying information. HLA-A*02:01+, HLA-B*57:01+leukapheresis samples (leukopaks) were obtained from StemCell. PBMCs were isolated by density-based centrifugation using Ficoll (GE Healthcare). CD14 +monocytes were isolated from PBMCs using a CD14 positive-selection kit (Stemcell). Isolated monocytes were differentiated in R10 media [RPMI 1640 without phenol red (Gibco) supplemented with 10% heat-inactivated FBS (Gibco), 1% HEPES (Corning), 1% L-glutamine (Sigma)] supplemented with 25 ng/mL M-CSF (Biolegend, 572902). Monocytes were cultured on appropriate ultra-low-attachment plates or flasks (Corning) for 6 days.

For ELISA assays, Cathepsin B activity assays, and growth curves (see below), after 6 days macrophages were detached using a detachment buffer of calcium-free PBS and 2 mM ethylenediaminetetraacetic acid (EDTA), pelleted, and recounted. Macrophages were plated in tissue culture treated 96-well plates at a density of 50,000 cells per well in R10 media and allowed to re-adhere overnight prior to infection. For microscopy (see below) cells were similarly detached and were replated on 12-well chamber slides (Ibidi) at a density of 100,000 cells per well in R10 media and allowed to re-adhere overnight prior to infection. For all other experiments, macrophage differentiation media was replaced with fresh R10 overnight prior to infection, without re-plating the macrophages.

### HLA genotyping

Genomic DNA was extracted from $5x10^6$ PBMCs using a Qiagen DNeasy kit. HLA typing was performed using a targeted next generation sequencing (NGS) method. Briefly, locus-specific primers were used to amplify a total of 26 polymorphic exons of HLA-A & B (exons 1–4), C (exons 1–5), E (exon 3), DPA1 (exon 2), DPB1 (exons 2–4), DQA1 (exon 1–3), DQB1 (exons 2 & 3), DRB1 (exons 2 & 3), and DRB3/4/5 (exon 2) genes with Fluidigm Access Array system (Fluidigm Corporation, South San Francisco, CA 94080 USA). The 26 Fluidigm PCR amplicons were harvested from Fluidigm Access Allay IFC and pooled. Quality and quantity were checked using a Caliper LabChip GX Touch HT Nucleic Acid Analyzer (PerkinElmer, Waltham, MA 02452 USA). The PCR product library was quantitated and subjected to sequencing on an Illumina MiSeq sequencer (Illumina, San Diego, CA 92122 USA). HLA alleles and genotypes were called using the Omixon HLA Explore (version 2.0.0) software (Omixon Biocomputing Ltd., Budapest, Hungary).

### *Mtb* culture

*Mycobacterium tuberculosis* (*Mtb*) H37Rv was grown in Difco Middlebrook 7H9 media supplemented with 10% OADC, 0.2% glycerol, and 0.05% Tween-80 to mid-log phase. Strains expressing GFP were grown in media supplemented with 50 µg/mL hygromycin B (Sigma-Aldrich), and strains expressing *luxABCDE* were grown in media supplemented with 20 µg/mL Zeocin (Thermo).

### *Mtb* infection of macrophages

The *Mtb* culture was pelleted by centrifugation, washed once with PBS, resuspended in R10 media and filtered through a 5 µm syringe filter to obtain a single-cell suspension. Macrophages were infected at the indicated multiplicity of infection (MOI) for 4 hr and then washed with PBS to remove extracellular *Mtb*. Infected macrophages were cultured in R10 media for the remainder of the experiment.

## Drug treatment of macrophages

Where indicated, macrophages were pre-treated with R10 media containing 20 µM E64d (Cayman), 0.5 µM MG-132 (Cayman), or 10 nM bafilomycin (InvivoGen), or DMSO alone (or other concentration of drug where indicated). The culture media was supplemented with an equal concentration of drug during infection with *Mtb*, and after washing off extracellular *Mtb*. Where indicated, the culture media was supplemented with 10 ng/mL of IFN-β after infection with *Mtb*.

## hipMHC internal standard preparation

UV-mediated peptide exchange and quantification of hipMHC complexes by ELISA were performed as previously described (*Stopfer et al., 2020*). Amidated peptides with sequences AL[+7]ADGVQKV-NH$_2$, ALNEQIARL[+7]-NH$_2$, and SLPEEIGHL[+7]-NH$_2$ were used to make hipMHC standards.

## MHC-I immunoprecipitation (IP)

A total of 50 million macrophages per condition (data-dependent analysis) or 10 million macrophages per condition (quantitative SureQuant analysis) were cultured as described above in ultra-low-attachment T75 flasks (Corning). macrophages were infected at MOI 2.5 as described above, or mock-infected with media containing no *Mtb*. Where indicated for specific DDA MS experiments (donor B), the culture media was supplemented with 10 ng/mL of IFN-γ for 24 hr prior to infection with *Mtb*. Where indicated for specific DDA MS experiments (donor C), the culture media was supplemented with 0.5 µg/mL of cycloheximide for the final 6 hr of culture prior to MHC-I isolation. Seventy-two hr post-infection, cells were detached using PBS supplemented with 2 mM EDTA, washed with PBS, and lysed in 1 mL (DDA analysis) or 0.5 mL (SureQuant analysis) of MHC lysis buffer [20 mM Tris, 150 mM sodium chloride, pH 8.0, supplemented with 1% CHAPS, 1 x HALT protease and phosphatase inhibitor cocktail (Pierce), and 0.2 mM phenylmethylsulfonyl fluoride (Sigma-Aldrich)]. Lysate was sonicated using a Q500 ultrasonic bath sonicator (Qsonica) in five 30 s pulses at an amplitude of 60%, cleared by centrifugation at 16,000 x *g*, and sterile filtered twice using 0.2 µm filter cartridges (Pall NanoSep). For quantitative SureQuant analyses, the protein concentration of the lysates was normalized by BCA assay (Pierce), and 100 fmol of each hipMHC standard was added to each sample. Lysates were then added to protein A sepharose beads pre-conjugated to pan-MHC-I antibody (clone W6/32), prepared as previously described (*Stopfer et al., 2020*), and incubated rotating at 4 °C overnight (12–14 hours). Beads were then washed and peptide-MHC complexes eluted as previously described (*Stopfer et al., 2020*).

## Purification of MHC-I-associated peptides

### Protocol 1

MHC-I-associated peptides were purified using 10 kDa molecular weight cutoff filters (Pall NanoSep) as previously described (*Stopfer et al., 2020*), snap-frozen in liquid nitrogen, and lyophilized.

### Protocol 2

C18 SpinTips (Pierce) were washed with 0.1% trifluoroacetic acid, activated with 90% acetonitrile supplemented with 0.1% formic acid, and washed with 0.1% formic acid. Eluate from MHC-I IPs was applied to the column by centrifugation. The column was washed with 0.1% formic acid, and peptides were eluted by applying elution solvent (30% acetonitrile supplemented with 0.1% formic acid) by centrifugation twice. Eluates were snap-frozen in liquid nitrogen and lyophilized. Lyophilized peptides were resuspended in solvent A2 (10 mM triethyl ammonium bicarbonate [TEAB], pH 8.0) and loaded onto a fractionation column (a 200 µm inner diameter fused silica capillary packed in-house with 10 cm of 10 µM C18 beads). Peptides were fractionated using an Agilent 1100 series liquid chromatograph using buffers A2 (10 mM TEAB, pH 8.0) and B2 (99% acetonitrile, 10 mM TEAB, pH 8.0). The fractionation column was washed with solvent A2, and peptides were separated using a gradient of 1–5% solvent B2 over 5 min, 5–40% over 60 min, 40–70% over 10 min, hold for 9 min, and 70%–1% over 1 min. Ninety-second fractions were collected, concatenated into 12 tubes, over 90 min in total. Fractions were flash-frozen in liquid nitrogen and lyophilized.

## DDA MS analyses

MHC-I peptide samples were resuspended in 0.1% formic acid. 25% of the sample was refrozen and reserved for later SureQuant validation analyses, while 75% of the sample was used for DDA analysis.

For all MS analyses, samples were analyzed using an Orbitrap Exploris 480 mass spectrometer (Thermo Fisher Scientific) coupled with an UltiMate 3000 RSLC Nano LC system (Dionex), Nanospray Flex ion source (Thermo Fisher Scientific), and column oven heater (Sonation). The MHC peptide sample was loaded onto a fused silica capillary chromatography column with an integrated electrospray tip (~1 µm orifice) prepared and packed in-house with 10 cm of 1.9 µm C18 beads (ReproSil-Pur).

Standard mass spectrometry parameters were as follows: spray voltage, 2.0 kV; no sheath or auxiliary gas flow; ion transfer tube temperature, 275 °C. The Orbitrap Exploris 480 mass spectrometer was operated in data dependent acquisition (DDA) mode. Peptides were eluted using a gradient of 6–25% buffer B (70% Acetonitrile, 0.1% formic acid) over 75 min, 25–45% over 5 min, 45–100% over 5 min, hold for 1 min, and 100% to 3% over 2 min. Full scan mass spectra (350–1200 m/z, 60,000 resolution) were detected in the orbitrap analyzer after accumulation of $3\times10^6$ ions (normalized AGC target of 300%) or 25ms. For every full scan, MS/MS scans were collected during a 3 s cycle time. Ions were isolated (0.4 m/z isolation width) using the standard AGC target and automatic determination of maximum injection time, fragmented by HCD with 30% CE, and scanned at a resolution of 45,000. Charge states <2 and>4 were excluded, and precursors were excluded from selection for 30 s if fragmented n=2 times within a 20-s window.

## DDA data search and manual inspection

All mass spectra were analyzed with Proteome Discoverer (PD, version 2.5) and searched using Mascot (version 2.4) against a custom database comprising the Uniprot human proteome (UP000005640) and the Uniprot *Mycobacterium tuberculosis* H37Rv proteome (UP000001584). No enzyme was used, and variable modifications included oxidized methionine for all analyses. Peptide-spectrum matches from all analyses were filtered with the following criteria: search engine rank = 1, isolation interference ≤30%, length between 8 and 16 amino acids, ions score ≥15, and percolator q-value <0.05. Identifications (IDs) of putative *Mtb*-derived peptides were rejected if any peptide-spectrum matches (PSMs) for the same peptide were found in the unfiltered DDA MS data for the corresponding mock-infected control. For each putative *Mtb* peptide identified, MS/MS spectra and extracted ion chromatograms (XIC) were manually inspected, and the ID was only accepted for further validation if it met the following criteria: (1) MS/MS spectra contained enough information to unambiguously assign a majority of the peptide sequence; (2) neutral losses were consistent with the chemical properties of the peptide; (3) manual de novo sequencing did not reveal an alternate peptide sequence that would explain a greater number of MS/MS spectrum peaks; (4) XIC showed a peak in MS intensity at the mass to charge ratio (m/z) of the peptide precursor ion at the retention time at which it was identified that did not appear in the corresponding mock-infected control. Peptides that met these criteria were further validated using SureQuant (see below).

DDA MS data were searched a second time against a database comprising only known T7SS substrates of *Mtb* H37Rv (all Esx-family proteins, all PE/PPE proteins, and EspA, EspB, EspC, EspE, EspF, EspI, EspJ, and EspK). Because high sequence similarity among many T7SS substrates artificially raises percolator q-values computed using this limited database, these searches were filtered only on ions score ≥20 rather than on ions score ≥15 and percolator q-value <0.05. The only validated *Mtb* peptide identified via these additional searches that had not previously been identified (AEHGMPGVP, derived from PPE51) was omitted from enrichment analyses (see below) to avoid introducing bias.

## Gibbs clustering

Host peptides and putative *Mtb* peptides that passed the filters described above were clustered using GibbsCluster 2.0 (*Andreatta et al., 2017*), hosted at (https://services.healthtech.dtu.dk/service.php?GibbsCluster-2.0). Default parameters for MHC-I peptide clustering were used, with the modification that a maximum of 6 clusters was allowed. The number of clusters that gave the highest KL divergence score was considered optimal. Clusters were assigned to HLA alleles by comparing cluster motifs to the known sequence binding motifs obtained from the NetMHCpan 4.1 motif viewer (*Reynisson et al., 2020*) (https://services.healthtech.dtu.dk/service.php?NetMHCpan-4.1) for HLA alleles expressed by that donor.

## HLA allele assignment

For each donor, the binding affinity of *Mtb* peptides identified in DDA MS data for each HLA allele expressed by that donor (as determined by HLA typing – see above) was predicted using NetMHCpan 4.1 (*Reynisson et al., 2020*). Peptides were assigned to the allele for which they had the highest predicted binding score.

## Enrichment analyses

To determine whether *Mtb*-derived MHC-I peptides or their source proteins were enriched for proteins with specific secretion signals, we classified each protein in the reference *Mtb* H37Rv proteome (UP000001584) using SignalP 6.0 (*Teufel et al., 2022*), and added an additional class of known T7SS substrates (as defined above; see DDA data search). Under a 'neutral model' in which all possible *Mtb* peptides are equally likely to be presented, the probability of drawing a peptide from a given class of proteins can be approximated as the sum of the lengths of the amino acid sequences of proteins in that class, divided by the total number of amino acid residues in the *Mtb* proteome. Using this approximation, we used a binomial test to assess whether the number of *Mtb*-derived peptides from each class of proteins detected in the MHC-I was greater or less than would be predicted under the neutral model. We used a hypergeometric test to assess whether the number of source proteins from each class was greater or less than expected, relative to a neutral model in which each protein in the *Mtb* proteome is equally likely to contribute to the MHC-I repertoire.

## SIL peptide synthesis

SIL peptides were synthesized at the MIT Biopolymers and Proteomics Lab using standard Fmoc chemistry using an Intavis model MultiPep peptide synthesizer with HATU activation and 5 μmol chemistry cycles. Starting resin used was Fmoc-Amide Resin (Applied Biosystems). Cleavage from resin and simultaneous amino acid side chain deprotection was accomplished using: trifluoroacetic acid (81.5% v/v); phenol (5% v/v); water (5% v/v); thioanisole (5% v/v); 1,2-ethanedithiol (2.5% v/v); 1% triisopropylsilane for 4 hr. Standard Fmoc amino acids were procured from NovaBiochem and SIL Fmoc-Leu ($^{13}C_6$, $^{15}N$), Fmoc-Arg ($^{13}C_6$), Fmoc-Glu ($^{13}C_5$ $^{15}N$) were obtained from Cambridge Isotope Laboratories. Peptides were quality controlled by MSy and reverse phase chromatography using a Bruker MicroFlex MALDI-TOF and Agilent model 1100 HPLC system with a Vydac C18 column (300 Å, 5 μm, 2.1×150 mm$^2$) at 300 μL/min monitoring at 210 and 280 nm with a trifluoroacetic acid/H$_2$O/MeCN mobile phase survey gradient.

## Synthetic standard survey MS analyses

DDA MS analysis of the SIL peptide mixture was performed as described above (see DDA MS analyses) with the following modifications: Peptides were eluted using a gradient of 6–35% buffer B over 30 min, 35–45% over 2 min, 45–100% over 3 min, and 100% to 2% over 1 min. No dynamic exclusion was used.

A second set of survey analyses was performed on the mixture of SIL peptides with background matrix using the full SureQuant acquisition method (see below). SIL peptides were spiked into a mixture of MHC-I peptides purified as described above from THP-1 cells differentiated into macrophages via 24 hr of treatment with 150 nM phorbol myristate acetate (PMA), which provided a representative background matrix. Because SIL amino acids are not 100% pure, SIL peptide concentrations were adjusted and survey analyses were repeated until the SIL peptide could be reliably detected while minimizing background signal detected at the mass of the biological peptide.

## SureQuant MS analyses
### Validation analyses

Standard MS parameters and MS1 scan parameters were as described above (see DDA MS analyses). The custom SureQuant acquisition template available in Thermo Orbitrap Exploris Series 2.0 was used to build four methods for validation analyses. One method targeted peptides detected in DDA analyses of the MHC-I repertoire of macrophages from donors A, B, and C. The other three methods targeted peptides detected in DDA analyses of the MHC-I repertoire of macrophages from donors D, E, and F, respectively. For each method, after the optimal charge state and most intense product ions were determined via a survey analysis of the synthetic SIL peptide standards (see above), one method

branch was created for each m/z offset between the SIL peptide and biological peptide as previously described (*Stopfer et al., 2020*). A threshold of n=3 out the top 6 product ions was used for pseudo-spectral matching, with a mass accuracy tolerance of 10 ppm.

## Quantitative analyses

Quantitative SureQuant analyses were performed similarly, with an additional method branch added to target hipMHC peptides using an inclusion list. As none of the targets of these analyses has an m/z<380, an MS1 scan range of m/z 380–1500 was used to exclude some common background ions. Targeted MS data were analyzed using Skyline Daily Build 22.1.9.208. For each target peptide, the intensities of the three most intense product ions were integrated over the time during which the peptide was scanned. These intensities were normalized by the integrated intensities of the corresponding product ions from the corresponding SIL standard over the same time interval, and these ratios were averaged. Finally, these averaged ratios were normalized by the average ratio of the integrated intensities of the hipMHC standard peptides and normalized to the DMSO-treated, *Mtb*-infected condition to obtain the final relative abundance of each target. 100 fmol of SIL EsxA$_{28-36}$ and 1 pmol of SIL EsxJKPW$_{24-34}$ were spiked into each sample per analysis, as well as (where indicated) 250 fmol of SIL standard for each of the target self peptides.

## Immunofluorescence microscopy

100,000 macrophages per well plated on chamber slides (Ibidi) were infected with GFP-expressing *Mtb* at MOI 1 or mock-infected with media containing no *Mtb*. At the indicated time point post-infection, macrophages were washed with PBS and fixed with 4% paraformaldehyde (PFA) in PBS for 1 hr. After the full time course was collected, all slides were blocked with 5% normal goat serum in PBS supplemented with 0.3% v/v Triton X-100 for 1 hr at room temperature and stained overnight at 4 °C with primary antibody (LAMP1 – Cell Signaling Technologies [CST] D2D11; P62 – CST D10E10; MHC-I – CST D8P1H; Galectin-3 – Thermo A3A12) diluted in antibody dilution buffer (PBS with 1% w/v bovine serum albumin and 0.3% v/v triton X-100). Wells were washed three times with PBS for 5 min each and stained with Alexa fluor 647 (AF647)-conjugated goat anti-rabbit or anti-mouse secondary antibody (Thermo) diluted to a final concentration of 1 μg/mL in antibody dilution buffer at room temperature for 2 hr. Wells were washed three times with PBS for 5 min each and stained with DAPI at a final concentration of 300 nM in PBS for 15 min at room temperature. Wells were washed three times with PBS for 5 min each, well dividers were removed, and slide covers were mounted on slides using Prolong Diamond anti-fade mounting media (Thermo). Slides were imaged using a TissueFAXS Confocal slide scanner system (TissueGnostics). 100 fields of view were acquired per condition using a ×40 objective lens.

## Segmentation of *Mtb*-containing phagosomes

Image segmentation was performed using opencv-python 4.5.4.60. A two-dimensional Gaussian blur with a standard deviation of five pixels was applied to de-noise GFP fluorescence images. A binary mask was generated from the blurred GFP image using a fluorescence intensity threshold that was empirically selected for each biological replicate. GFP +phagosomes were further segmented using the watershed algorithm.

## Colocalization analysis

A correlation image was generated by computing the Pearson correlation coefficient between the GFP fluorescent intensity and AF647 fluorescent intensity in a 41x41 pixel sliding window with a step size of 1. This correlation value was averaged over each *Mtb*-containing phagosome, and phagosomes with a mean value of greater than or equal to 0.6 were considered co-localized. For statistical analyses of log-transformed rates of co-localization, zeros were replaced with the limit of detection for a given sample (100% x [1 /N] where N is the number of *Mtb*-containing phagosomes detected).

## *Mtb* growth curves

A total of 50,000 macrophages per well were infected in technical triplicate in opaque white 96-well plates with *luxABCDE*-expressing *Mtb* at MOI 2.5 (or mock-infected with media not containing *Mtb*). Where indicated, macrophages were treated with drug as described above, or media was

supplemented with 10 ng/mL IFN-β (BioLegend) or 25 µg/mL rifampicin (RIF) after infection. Luminescence was measured at the indicated time points using a Spark 10 M plate reader (Tecan).

## Phagocytosis assay

1x10⁶ macrophages were differentiated as described above on six-well ultra low-attachment plates and infected with GFP-expressing *Mtb* at an MOI of 2.5 (or mock-infected with media containing no *Mtb*). After the 4 hr infection, macrophages were detached using PBS supplemented with 2 mM EDTA, washed with PBS, stained with Live/Dead Fixable Near-IR dye (Thermo) for 10 min, washed with FACS buffer (PBS supplemented with 2% FBS and 2 mM EDTA) and PBS, fixed with PBS containing 4% PFA for one hour, and washed and resuspended in FACS buffer for analysis on an LSRFortessa flow cytometer (BD).

## MHC-I and MHC-II flow cytometry

A total of 1x10⁶ macrophages were differentiated as described above on six-well ultra low-attachment plates and infected with GFP-expressing *Mtb* at an MOI of 2.5 (or mock-infected with media containing no *Mtb*). Seventy-two hr post-infection, macrophages were detached using PBS supplemented with 2 mM EDTA, blocked with Human TruStain FcX Fc receptor blocking solution (BioLegend) for 10 min, stained with phycoerythrin (PE)-conjugated anti-HLA-A,B,C antibody (BioLegend, clone W6/32) or PE-conjugated anti-HLA-DQ antibody (BioLegend, clone HLA-DQ1) and PerCP-Cy5.5-conjugated anti-HLA-DR antibody (BioLegend, clone L243) for 20 min, stained with Live/Dead Fixable Near-IR dye (Thermo) for 10 min, washed with FACS buffer and PBS, fixed with PBS containing 4% PFA for 1 hr, and washed and resuspended in FACS buffer for analysis on an LSR Fortessa flow cytometer (BD).

## ELISA

A total of 50,000 macrophages per well were infected in technical triplicate in a 96-well plate format with *Mtb* at MOI 2.5 (or mock-infected with media not containing *Mtb*). Seventy-two hr post-infection, culture media was collected and sterile filtered twice by centrifugation using 96-well 0.2 µm filter plates (Corning). The concentration of CXCL10 in the media was determined using a Human CXCL10 ELISA MAX kit (BioLegend 439904).

## LDH release assay

A total of 50,000 macrophages per well were infected in technical triplicate in a 96-well plate format with *Mtb* at MOI 2.5 (or mock-infected with media not containing *Mtb*) and treated with the indicated concentrations of drug or vehicle control. Twenty-four hr post-infection, culture media was collected and sterile filtered twice by centrifugation using 96-well 0.2 µm filter plates (Corning). Percent cytotoxicity was determined using a CyQuant LDH Cytotoxicity Assay kit (Thermo C20300).

## Western blots

On six-well ultra low-attachment plates, 1x10⁶ macrophages were differentiated as described above . Where indicated, macrophages were infected with *Mtb* at an MOI of 2.5. Macrophages were treated for 24 hr with the indicated concentration of MG-132 or DMSO control, and were then detached using PBS supplemented with 2 mM EDTA, washed with PBS, pelleted, and lysed in 100 µL RIPA buffer. Lysates were cleared by centrifugation at 16,000 x *g* for 5 min. For cells infected with *Mtb*, lysates were filtered twice using 96-well 0.2 µm filter plates (Corning). NuPAGE LDS Sample Buffer (Thermo) and NuPAGE Sample Reducing Agent (Thermo) were added to a final concentration of 1 x, and 20 µL of lysate per well was separated by SDS-PAGE. Protein was transferred to a PVDF membrane using an iBlot2 dry transfer system (Thermo). Membranes were blocked for 1 hr at room temperature with blocking buffer (Licor) and incubated overnight at 4 °C with anti-K48-linked poly-ubiquitin antibody (Cell Signaling Technologies) and anti-β-actin antibody (Santa Cruz Biotechnologies), each diluted 1:1000 in antibody diluent (Licor). Blots were washed three times for 5 min each with tris-buffered saline with 0.1% tween-20 (TBS-T) and incubated for 1 hr at room temperature with IRDye 800CW goat anti-rabbit secondary antibody (Licor) and IRDye 680CW goat anti-mouse secondary antibody (Licor), each diluted 1:10,000 in antibody diluent (Licor). Blots were washed three times for five minutes each with TBS-T and imaged using an Odyssey DLx imaging system (Licor).

## Cathepsin B activity assays

Cathepsin B activity was measured using an assay protocol adapted from prior literature (*Creasy et al., 2007*; *Hulkower et al., 2000*). 50,000 macrophages per well were infected in technical triplicate in a 96-well plate format with *Mtb* at MOI 2.5 (or mock-infected with media not containing *Mtb*) and treated with the indicated concentrations of E64d or vehicle control. Twenty-four hr post-infection, macrophages were washed with PBS and lysed by adding 100 μL assay buffer (87.7 mM $KH_2PO_4$, 12.3 mM $NaHPO_4$, 4 mM EDTA, pH 5.5) supplemented with 2.6 mM dithiothreitol and 0.1% v/v Triton X-100, mixing thoroughly, and incubating at 37 °C for 15 min. Forty μL of lysate was transferred to a separate 96-well plate, and the fluorogenic cathepsin B substrate Z-RR-AMC (Sigma-Aldrich) diluted in assay buffer was added to a final concentration of 2 mM. Plates were immediately transferred to a Spectramax iD3 plate reader (Molecular Devices) pre-warmed to 37 °C. Fluorescence at an absorbance wavelength of 370 nm and emission wavelength of 460 nm was measured in kinetic read mode at intervals of 2 min for a total duration of 20 min. The linear (enzyme-limited) regime of the fluorescence curve was determined to last approximately 10 min, so cathepsin B activity was quantified by using linear regression to determine the initial rate of fluorescence increase over the first 10 min of the assay.

## Lysotracker microscopy

A total of 100,000 macrophages per well plated on chamber slides (Ibidi) were infected with GFP-expressing *Mtb* at MOI 2.5 (or mock-infected with media containing no *Mtb*) and treated with bafilomycin or vehicle control as described above. Twenty-four hr post-infection, macrophages were washed with Hank's Balanced Salt Solution (HBSS), then incubated for 5 min at 37 °C in pre-warmed HBSS containing 1 μM Lysotracker Red (Thermo). Cells were washed twice with HBSS and fixed with PBS containing 4% PFA for 1 hr, then stained with DAPI and mounted as described above (see immunofluorescence microscopy). Slides were imaged using a TissueFAXS Confocal slide scanner system (TissueGnostics). Using a ×40 objective lens, 100 fields of view were acquired per condition.

## Automated counting of nuclei

Image segmentation was performed using opencv-python 4.5.4.60. A two-dimensional Gaussian blur with a standard deviation of five pixels was applied to de-noise DAPI fluorescence images. A binary mask was generated from the smoothed DAPI image using a fluorescence intensity threshold that was determined automatically using Otsu's thresholding method, as implemented in opencv-python. To disambiguate adjacent or overlapping nuclei, nuclei were further segmented using the watershed algorithm.

## Calculation of Lysotracker+ area per cell

A two-dimensional Gaussian blur with a standard deviation of five pixels was applied to de-noise Lysotracker Red fluorescence images. A binary mask was generated from the Lysotracker Red image using a fixed intensity threshold for all samples. The total Lysotracker +area was computed from the binary mask and divided by the number of nuclei counted in the image (see above).

## Curve fitting and statistical analyses

Unless otherwise indicated, all nonlinear curve fitting and all statistical tests were performed in GraphPad Prism 9.

## Acknowledgements

The authors thank all the staff members of the Ragon Institute, the Koch Institute, and MIT for the essential work they do to make our research possible. The authors thank Lauren Stopfer, Akeem Ngomu Akilimali, Sean Kim, Elizabeth Choe, Charul Jani, Yong Xie (Ragon Institute BSL3 Facility), Thomas Diefenbach (Ragon Institute Microscopy Core Facility), Tigist Tamir, Cameron Flower, Angela Ahn, Kiera Clayton, and Aaron Shulkin for helpful conversations, training, and technical guidance. Iris Abrahantes Morales isolated monocytes from buffy coats. The authors thank Yuko Yuki and Mary Carrington (National Cancer Institute) for HLA genotyping of primary human cells. Kiera Clayton generously shared HLA-A*02:01+, HLA-B*57:01+PBMCs from a leukapheresis sample for one biological

replicate. The *eccCa1:Tn Mtb* strains used in this study were generously provided by Charul Jani and Amy Barczak. Diane Ballestas and Isadora Deese provided administrative support. Alla Leshinsky, Heather Amoroso, and Richard Cook (Koch Institute Biopolymers Core Facility) synthesized and purified SIL peptide standards. We modified code written by Cameron Flower to plot MS/MS spectra. The authors thank the staff of the Ragon Institute Flow Cytometry Core Facility. This work was supported by funding from NIH grants R35GM142900-01 and R01AI022553, NIEHS grant P42 ES027707, and the Center for Precision Cancer Medicine. This work was performed in part in the Ragon Institute BSL3 core, which is supported by the NIH-funded Harvard University Center for AIDS Research (P30 AI060354).

## Additional information

### Funding

| Funder | Grant reference number | Author |
| --- | --- | --- |
| National Institutes of Health | R35GM142900-01 | Owen Leddy |
| National Institutes of Health | R01AI022553 | Owen Leddy |
| National Institute of Environmental Health Sciences | P42 ES027707 | Owen Leddy Forest M White |
| Center for Precision Cancer Medicine | | Owen Leddy Forest M White |
| National Institutes of Health | P30 AI06035 | Owen Leddy |

The funders had no role in study design, data collection and interpretation, or the decision to submit the work for publication.

### Author contributions

Owen Leddy, Conceptualization, Data curation, Software, Formal analysis, Validation, Investigation, Visualization, Methodology, Writing – original draft, Project administration, Writing – review and editing; Forest M White, Bryan D Bryson, Conceptualization, Data curation, Supervision, Funding acquisition, Investigation, Methodology, Writing – original draft, Project administration, Writing – review and editing

### Author ORCIDs

Owen Leddy ⬥ https://orcid.org/0000-0002-3437-1956
Forest M White ⬥ https://orcid.org/0000-0002-1545-1651
Bryan D Bryson ⬥ https://orcid.org/0000-0003-1716-6712

### Decision letter and Author response

Decision letter https://doi.org/10.7554/eLife.84070.sa1
Author response https://doi.org/10.7554/eLife.84070.sa2

## Additional files

### Supplementary files

MDAR checklist

### Data availability

The mass spectrometry data have been deposited to the ProteomeXchange Consortium via the PRIDE (*Perez-Riverol et al., 2022*) partner repository with the dataset identifiers PXD037837 (DDA data) and PXD037843 (SureQuant data). Microscopy data analysis scripts are available on GitHub at https://github.com/oleddy/local_correlation_analysis (copy archived at *Leddy, 2023*).

The following datasets were generated:

| Author(s) | Year | Dataset title | Dataset URL | Database and Identifier |
|---|---|---|---|---|
| Bryson BD | 2023 | DDA data | https://www.ebi.ac.uk/pride/archive/projects/PXD037837 | PRIDE, PXD037837 |
| Bryson BD | 2023 | SureQuant data | https://www.ebi.ac.uk/pride/archive/projects/PXD037843 | PRIDE, PXD037843 |

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
