## [Editor Report]

This landmark study uses compelling approaches such as quantitative and screening mass spectrometry to identify peptides from tuberculosis bacteria that are presented by macrophages infected with this pathogen. The authors provide convincing evidence that the presentation of these antigens depends on a specialist bacterial secretion system. The study will be of interest to infectious disease specialists and of particular value for future vaccine development.

---

## [Decision Letter]

**Decision letter after peer review:**

Thank you for submitting your article "Immunopeptidomics reveals determinants of *Mycobacterium tuberculosis* antigen presentation on MHC class I" for consideration by *eLife*. Your article has been reviewed by 3 peer reviewers, and the evaluation has been overseen by a Reviewing Editor and Bavesh Kana as the Senior Editor. The reviewers have opted to remain anonymous.

Essential revisions:

1. Could the authors speculate why certain antigens (e.g. Rv0288, Rv1886c) that were previously found to be HLA*A2 restricted and elicit abundant Class-I restricted CD8 T cell responses in humans (Axelsson-Robertson, Int J Inf Dis, 2015), were not found to be presented by infected macrophages in this study? It would be useful to cite and discuss this point.

2. Although the authors used macrophages from donors expressing HLA-B*57:01 and HLA-A*02:01, other HLA alleles are expressed on these macrophages. Furthermore, other Mtb peptides were also likely to be presented (such as the PE-PPE proteins, those associated with the mycobacterial outer membrane, or sec substrate TB8.4 mentioned in lines 318-320). Were other peptides found in these quantitative MS experiments? How did their repertoire compare to those found in the initial experiments (even though the focus was on quantifying the 2 peptides mentioned)? If data are available, they would strengthen the results, if not, this should be discussed.

Related: Were peptides other than the 2 that were quantified affected by the Esx1 deletion mutant? (ie non-ESX proteins)

3. Bafilomycin and E64D appear to increase the relative abundance of both the Mtb peptides and other self-peptides. Does the use of these agents also correlate with a shift in MHC-I vs. MHC-II expression on the cell surface? Perhaps related to the mass effect if the Class-II pathway is stalled due to impaired phagolysosomal fusion.

4. If the peptides presented on MHC-I are not affected by proteasome or cathepsin B inhibition, it would be helpful if the authors could discuss why that is the case and which processes are likely responsible for antigen processing in this case (in addition to what is written in lines 252-255).

5. There are a few limitations of this study that should be addressed in the discussion. First, please address in the text why H37Rv was chosen- this is a virulent lab strain but perhaps not the best choice, as it is known to be less virulent than other strains like Erdman, CDC1551 or the Beijing strains. Adding something about this choice and why it might be a limitation of the study would be worth the author's time. Second, the suggestion that ESX-1 is required for the presentation of the antigens from other ESX systems is based on the quantification of two peptides (one from ESX-1 and one from Esx3/5). This may be a limitation of the current study. If it is not, please discuss.

Other points that should be addressed:

1. The authors should mention the rationale/significance of infecting the macrophages for 72h (rather than 24-48h) in the first set of immunopeptidomics experiments. Mtb infection >48h has been associated with the downregulation of MHC-II in infected cells.

2. Hypotheses 1 and 2 are presented in the opposite order in the text (lines 133-134) as in Figure 2a.

3. Citation for the statement in line 292

4. It would be helpful if the rationale for (and the difference between) the Esx5 peptides QTVEDEARRMW vs. QTVEDEARRM(ox)W were mentioned in the Results section in addition to the legend for Figure 3. I had to search to understand why only 2 peptides were mentioned in the text, but data for 3 peptides were shown in the figure.

5. MG-132 increased the expression of extracellular and endosomal peptides, but reduced Mtb and nuclear/cytosolic proteins, which is internally consistent and strengthens the authors' conclusions that the inhibitor worked, but did not affect the expression of the two Mtb peptides quantified.

6. It is unclear why the authors think that Rv1211 and Rv3196A are candidates for proteins that facilitate protein transport across the phagosomal membrane. This seems to be speculation and completely unsupported. (lines 312)

7. There are a few typos in the figure legends (sometimes panels are bolded, and sometimes they are not), and sometimes the order of the figure doesn't align with the figure legends.

8. The manuscript is lacking many of the key citations required to put this work into context. To strengthen this study, the authors should review, cite and discuss their work in the context of the following citations, to provide a clear understanding of what this study adds, and how this study moves the field forward.

Some of the references for the Type VII systems need to be altered. None of the references are the first report of the secretion systems. For ESX-1 please cite PMID: 14557536, for Esx-5 please cite PMID: 17076665, and for ESX-3 please cite PMID: 19846780. (lines 92-96)

The references for cytoplasmic access need revision. Lines 140-121; Please cite PMID: 17604718, for cytoplasmic access; PMID: 22319448, as well as Watson 2012 for ESX-1 dependency.

The authors need to consider the following references regarding ESX-1 and MHC presentation: PMID: 22022257, which suggests that access to the cytoplasm is required for MHC type 1 presentation in *M. tuberculosis*; PMID: 18591224, which shows that ESX-1 function is required for priming CD8 T cells, and presenting EsxB peptides, also found here. PMID: 19360129, defining the pathway used by MHC during M. tb infection for phagosomal antigens, PMID: 22901810, showed ESX-1 dependent localization of p62; It is already known that MTb localizes with markers of phagosomal membrane damage in an ESX-1 dependent manner.

The authors should consider these references demonstrating that BCG lacks ESX-1 activity (lines 288-290) References demonstrating BCG is attenuated because it lacks ESX-1: PMID: 10320585, BCG is missing RD1, PMID: 12410828 RD1 contributes to BCG attenuation, PMID: 12508154 deletion of RD1 from TB leads to attenuation, PMID: 12692540 restoring ESAT-6 secretion to BCG.

The authors should review these references (lines 300-302) first suggesting the model that the ESX-1 function precedes that of the other ESX systems (2-5). PMID: 33653883 first suggested that ESX-1 functions to lyse the phagosome, and that ESX-1 activity is followed by the activity of the other ESX secretion systems (see the final model in this paper). ESX-3 was first shown to function downstream of ESX-1 in this paper (PMID: 30482832).

The original papers reporting these proteins are secreted/ localized should be cited. PMID: 19682254 EspC was first reported as an ESX substrate here. For the list of proteins in 318-320. Please address the rest.

In lines 308-315, did the authors check if other papers that have defined the ESX-1 secretome have seen these proteins? Check PMID: 35671426 for the M. marinum orthologs.

*Reviewer #1 (Recommendations for the authors):*

The inclusion of T-cell activation experiments to validate the immunogenicity of some of the peptides would enhance the manuscript.

*Reviewer #2 (Recommendations for the authors):*

This is a very nice study. Overall, it is clever and well-presented and adds to the knowledge in the field. The approaches are rigorous, and the identification of mycobacterial peptides in the background of the host peptides is impressive. The conclusions are supported by the data presented. The figures and description of the methods are clear and transparent and will allow the field to reproduce this work. The authors could improve this manuscript by considering the points below. A major issue with the manuscript is that the references need to be reconsidered and improved. The study needs to be put into a clear context of the MHC, and ESX fields. We hope that this list of suggested references below will provide the authors with a means to do so.

The manuscript is lacking many of the key citations required to put this work into context. To strengthen this study, the authors should review, cite and discuss their work in the context of the following citations, to provide a clear understanding of what this study adds, and how this study moves the field forward.

Some of the references for the Type VII systems need to be altered. None of the references are the first report of the secretion systems. For ESX-1 please cite PMID: 14557536, for Esx-5 please cite PMID: 17076665, and for ESX-3 please cite PMID: 19846780. (lines 92-96)

The references for cytoplasmic access need revision. Lines 140-121; Please cite PMID: 17604718, for cytoplasmic access; PMID: 22319448, as well as Watson 2012 for ESX-1 dependency.

The authors need to consider the following references regarding ESX-1 and MHC presentation: PMID: 22022257, which suggests that access to the cytoplasm is required for MHC type 1 presentation in *M. tuberculosis*; PMID: 18591224, which shows that ESX-1 function is required for priming CD8 T cells, and presenting EsxB peptides, also found here. PMID: 19360129, defining the pathway used by MHC during M. tb infection for phagosomal antigens, PMID: 22901810, showed ESX-1 dependent localization of p62; It is already known that MTb localizes with markers of phagosomal membrane damage in an ESX-1 dependent manner.

The authors should consider these references demonstrating that BCG lacks ESX-1 activity (lines 288-290) References demonstrating BCG is attenuated because it lacks ESX-1: PMID: 10320585, BCG is missing RD1, PMID: 12410828 RD1 contributes to BCG attenuation, PMID: 12508154 deletion of RD1 from TB leads to attenuation, PMID: 12692540 restoring ESAT-6 secretion to BCG.

The authors should review these references (lines 300-302) first suggesting the model that the ESX-1 function precedes that of the other ESX systems (2-5). PMID: 33653883 first suggested that ESX-1 functions to lyse the phagosome, and that ESX-1 activity is followed by the activity of the other ESX secretion systems (see the final model in this paper). ESX-3 was first shown to function downstream of ESX-1 in this paper (PMID: 30482832)

The original papers reporting these proteins are secreted/ localized should be cited. PMID: 19682254 EspC was first reported as an ESX substrate here. For the list of proteins in 318-320. Please address the rest.

In lines 308-315, did the authors check if other papers that have defined the ESX-1 secretome have seen these proteins? Check PMID: 35671426 for the M. marinum orthologs.

It is unclear why the authors think that Rv1211 and Rv3196A are candidates for proteins that facilitate protein transport across the phagosomal membrane. This seems to be speculation and completely unsupported. (lines 312)

There are a few limitations of this study that should be addressed in the discussion. First, please address in the text why H37Rv was chosen- this is a virulent lab strain but perhaps not the best choice, as it is known to be less virulent than other strains like Erdman, CDC1551 or the Beijing strains. This is not an issue for this reviewer but I suspect adding something about this choice and why it might be a limitation of the study would be worth the author's time. Second, the suggestion that ESX-1 is required for the presentation of the antigens from other ESX systems is based on the quantification of two peptides (one from ESX-1 and one from Esx3/5). This may be a limitation of the current study. If it is not, please discuss this in the discussion.

There are a few typos in the figure legends (sometimes panels are bolded, and sometimes they are not), and sometimes the order of the figure doesn't align with the figure legends.

*Reviewer #3 (Recommendations for the authors):*

– Could the authors speculate why certain antigens (e.g. Rv0288, Rv1886c) that were previously found to be HLA*A2 restricted and elicit abundant Class-I restricted CD8 T cell responses in humans (Axelsson-Robertson, Int J Inf Dis, 2015), were not found to be presented by infected macrophages in this study? It would be useful to cite and discuss this point.

– Although the authors used macrophages from donors expressing HLA-B*57:01 and HLA-A*02:01, other HLA alleles are expressed on these macrophages. Furthermore, other Mtb peptides were also likely to be presented (such as the PE-PPE proteins, those associated with the mycobacterial outer membrane, or sec substrate TB8.4 mentioned in lines 318-320). Were other peptides found in these quantitative MS experiments? How did their repertoire compare to those found in the initial experiments (even though the focus was on quantifying the 2 peptides mentioned)? If data are available they would strengthen the results, but at least could be discussed.

Related: Were peptides other than the 2 that were quantified affected by the Esx1 deletion mutant? (ie non-ESX proteins)

– Bafilomycin and E64D appear to increase the relative abundance of both the Mtb peptides and other self-peptides. Does the use of these agents also correlate with a shift in MHC-I vs. MHC-II expression on the cell surface? Perhaps related to the mass effect if the Class-II pathway is stalled due to impaired phagolysosomal fusion.

– If the peptides presented on MHC-I are not affected by proteasome or cathepsin B inhibition, it would be helpful if the authors could discuss in the Discussion why that is the case and which processes are likely responsible for antigen processing in this case (in addition to what is written in lines 252-255).

---

## [Author Response]

Essential revisions:1. Could the authors speculate why certain antigens (e.g. Rv0288, Rv1886c) that were previously found to be HLA*A2 restricted and elicit abundant Class-I restricted CD8 T cell responses in humans (Axelsson-Robertson, Int J Inf Dis, 2015), were not found to be presented by infected macrophages in this study? It would be useful to cite and discuss this point.

This is an important point, and we appreciate the reviewers bringing it up. In general, while identifying a peptide in the MHC-I repertoire by MS provides strong positive evidence for its presentation, the fact that a given epitope was not detected in our MS analyses should not be interpreted as conclusive evidence that it cannot be presented on MHC-I in *Mtb* infection. We’ve made an effort to emphasize that there are several reasons why previously described T cell epitopes may not have been detected in our MS analyses:

1. Other types of APCs besides macrophages can also become infected with *Mtb* or otherwise take up *Mtb* antigen and prime CD8^+^ T cells during *Mtb* infection. If other cell types such as dendritic cells differ in the antigen processing proteases or antigen presentation pathways they express, they may present different sets of epitopes that could prime T cell responses such as those observed by Axelsson-Robertson et al. that are specific for epitopes we did not detect.

2. Depending on their physical and chemical properties, some MHC-I peptides may be partially or completely lost during sample handling (for example, very hydrophobic peptides may be retained on the C18 SPE column – see PMID: 36194871), or may not ionize efficiently and therefore produce an MS signal below the limit of detection.

3. DDA MS analyses are inherently stochastic (as the instrument makes real-time decisions in each analysis about which MS peaks to select for fragmentation) and. MHC-I peptides that were not picked for fragmentation, either due to lower abundance compared to other simultaneously eluting peptides, or by chance, will not be identified.

We’ve added a summary of these possible reasons to the discussion text (lines 432-447):

“We did not detect any peptides derived from several antigens previously shown to stimulate CD8^+^ T cell responses in Mtb infection, including responses restricted to HLA alleles represented in our DDA MS analyses [for example, EsxH and FbpB (Axelsson-Robertson et al., 2015)]. This could be due to any of several biological and/or technical reasons. Given that the Mtb-derived MHC-I peptides we detected were enriched for T7SS substrates but not for substrates of the Sec or Tat secretion pathways, it’s possible that the subcellular localization of antigens secreted via these pathways does not enable efficient presentation on MHC-I in macrophages. In this case, other types of APCs that differ in their antigen processing and presentation capabilities could still present peptides from these antigens to prime the previously observed CD8^+^ T cell responses. The same could also be true of certain T7SS substrate antigens (such as EsxH) if T7SS substrates vary in their subcellular localization or if presentation of certain epitopes requires antigen processing pathways not active in macrophages. Some peptides may also have been missed because they were lost during sample handling, because they were below the limit of detection due to low abundance or ionization efficiency, or because they co-eluted with many more abundant peptides and were therefore not prioritized for acquisition in DDA analyses. Our results don’t rule out presentation of additional antigens besides those we detected.”

2. Although the authors used macrophages from donors expressing HLA-B*57:01 and HLA-A*02:01, other HLA alleles are expressed on these macrophages. Furthermore, other Mtb peptides were also likely to be presented (such as the PE-PPE proteins, those associated with the mycobacterial outer membrane, or sec substrate TB8.4 mentioned in lines 318-320). Were other peptides found in these quantitative MS experiments? How did their repertoire compare to those found in the initial experiments (even though the focus was on quantifying the 2 peptides mentioned)? If data are available, they would strengthen the results, if not, this should be discussed.Related: Were peptides other than the 2 that were quantified affected by the Esx1 deletion mutant? (ie non-ESX proteins)

We thank the reviewers for pointing out a lack of clarity in the manuscript about the nature of the data we get out of our targeted quantitative MS analyses. We’ve attempted to clarify that these analyses only target the specific peptides for which we spike in corresponding stable isotope labeled (SIL) synthetic standards – specifically LLDEGKQSL (from EsxA), QTVEDEARRMW (from EsxJKPW) and, in the drug treatment experiments in Figure 4, several self peptides. Detection of the synthetic standard triggers acquisition of the biological peptide, so other peptides are not detected. Given that the primary cells used for these quantitative experiments were chosen such that all three donors expressed HLA-A*02:01 and HLA-B*57:01 but did not have any other HLA alleles in common, none of the other peptides we identified would have been detectable in cells from all three donors, even if corresponding SIL standards had been added. It’s unlikely that we would have been able to readily obtain cells from three or more donors all sharing three or more HLA of the relevant alleles.

We also emphasize the advantages of this approach that we feel compensate for the narrow focus on only a few peptides, including reliable acquisition across experimental conditions and the low limit of detection, which decreases the required sample input and makes experiments with many experimental conditions tractable.

We introduce these points in lines 202-205, which now read as follows: “Because the other Mtb epitopes detected in our untargeted MS experiments are not expected to bind these HLA alleles, we only targeted EsxA28-36 and EsxJKPW24-34. While this targeted approach is limited in the number of epitopes it can detect, it enables reliable and accurate quantification of peptides across experimental conditions with low sample input.”

3. Bafilomycin and E64D appear to increase the relative abundance of both the Mtb peptides and other self-peptides. Does the use of these agents also correlate with a shift in MHC-I vs. MHC-II expression on the cell surface? Perhaps related to the mass effect if the Class-II pathway is stalled due to impaired phagolysosomal fusion.

The reviewer raises an interesting hypothesis that we hadn’t previously considered. To determine whether a tradeoff between antigen presentation on MHC-I vs. MHC-II could lead to an increase in presentation of certain MHC-I epitopes in response to E64d or bafilomycin treatment, we analyzed flow cytometry data we had previously collected measuring surface MHC-I levels in *Mtb*-infected macrophages treated with these drugs, and also stained *Mtb*-infected macrophages treated with E64d or bafilomycin for surface HLA-DR and HLA-DQ. We found that treatment with E64d led to a slight and not statistically significant decrease in MHC-I and MHC-II expression while the effect of bafilomycin treatment on MHC-I and MHC-II surface levels was minimal. This led us to conclude that a tradeoff between directing antigen to MHC-I vs. MHC-II processing pathways was unlikely to be a major contributor to the effects of drug treatment that we observed in our targeted MS experiments. To present these data, we’ve extended Figure 4 —figure supplement 3 to include two additional panels D and E that present the MHC-I data, and added Figure 4 —figure supplement 4 to present the MHC-II data.

We describe the results of our analysis of surface MHC-I on drug-treated cells on lines 272-275, which now read “All three drugs exhibited minimal cytotoxicity in macrophages at the doses used in our immunopeptidomic experiments (Figure 4 —figure supplement 2), and did not inhibit phagocytosis of Mtb or bacterial outgrowth (Figure 4 —figure supplement 3), and did not have a significant effect on surface MHC-I levels (Figure 4 —figure supplement 3).”

We describe the results of our analysis of surface MHC-II on E64d and bafilomycin-treated cells on lines 284-288, which read: “Cell surface levels of HLA-DR and HLA-DQ were not significantly affected by bafilomycin or E64d treatment (Figure 4 —figure supplement 4), suggesting that E64d and bafilomycin did not substantially stall MHC-II antigen presentation and their effects of on MHC-I presentation could not be explained as indirect effects of modulating antigen entry into the MHC-II antigen processing pathway.”

We’ve amended the Methods subsection on flow cytometry to include a description of the MHC-II-specific antibodies used (lines 743-752).

4. If the peptides presented on MHC-I are not affected by proteasome or cathepsin B inhibition, it would be helpful if the authors could discuss why that is the case and which processes are likely responsible for antigen processing in this case (in addition to what is written in lines 252-255).

We appreciate the reviewer’s interest in possible alternative pathways of antigen processing, and we are eager to explore these possibilities in future work. We have added a paragraph to the discussion outlining some possible alternative proteolytic pathways that have previously been proposed to contribute to antigen processing on MHCs that could be involved in processing *Mtb* antigens presented on MHC-I (lines 379-389).

“We showed that presentation of two Mtb-derived peptides on MHC-I was independent of the activity of proteases commonly associated with antigen processing (the proteasome and cysteine cathepsins), suggesting an alternate processing mechanism. Other proteases have previously been proposed to contribute to proteolytic processing of antigens for presentation on MHCs, including tripeptidyl peptidase II (Geier et al., 1999; Guil et al., 2006; Lázaro et al., 2015; York et al., 2006), nardilysin (Kessler et al., 2011), thimet oligopeptidase (Kessler et al., 2011), metalloproteinases (Lorente et al., 2012), and serine cathepsins such as Cathepsin G that are not inhibited by E64d (Burster et al., 2010). Some of these or other enzymes may be responsible for proteolytic processing of EsxA_28-36_ and EsxJKPW_24-34_, or may have a redundant role that can compensate for inhibition of conventional antigen processing pathways. Further work will be needed to determine which proteolytic pathways contribute to presentation of Mtb T7SS substrates on MHC-I.”

5. There are a few limitations of this study that should be addressed in the discussion. First, please address in the text why H37Rv was chosen- this is a virulent lab strain but perhaps not the best choice, as it is known to be less virulent than other strains like Erdman, CDC1551 or the Beijing strains. Adding something about this choice and why it might be a limitation of the study would be worth the author's time. Second, the suggestion that ESX-1 is required for the presentation of the antigens from other ESX systems is based on the quantification of two peptides (one from ESX-1 and one from Esx3/5). This may be a limitation of the current study. If it is not, please discuss.

We appreciate the reviewer’s interest in how antigen presentation might vary among *Mtb* isolates. This is a question we also think will be interesting and important to investigate to understand both the fundamental biology of antigen presentation in TB and how to ensure broad vaccine protection across *Mtb* strains. We’ve added a paragraph addressing this point to the discussion (lines 372-377), which reads as follows:

“In this study, we used strain H37Rv for the sake of consistency with other prior studies on antigen presentation on MHC-I and phagosome membrane damage in Mtb-infection, but antigen presentation may vary among Mtb isolates. Mtb isolates have been shown to differ in their level of ESX-1 activity (Solans et al., 2014), as well as carrying mutations in Esx-family proteins and other T7SS substrates (Saelens et al., 2022; Uplekar et al., 2011). Further studies across multiple isolates may therefore reveal differences in antigen presentation that could be relevant for the design of broadly protective vaccines.”

Our ability to generalize about the ESX-1-dependence of MHC-I antigen presentation in *Mtb* infection is indeed limited by the fact that our quantitative experiments targeted two specific epitopes. To clarify this point, we’ve added more language in the discussion acknowledging the possibility that the requirements for antigen presentation on MHC-I may vary among different *Mtb* antigens. We’ve revised lines 340-343 to read “While ESX-1 activity is required for priming of CD8^+^ T cell responses specific for EsxB in vivo in mice, it is dispensable for priming of CD8^+^ T cell responses specific for antigens exported via other secretion systems (such as TB8.4 and EsxH) in vivo in mice (Woodworth et al., 2008a).” to emphasize the possibility of differences among antigens, and added lines 368-370, which read “Given that our quantitative MS experiments only targeted two Mtb-derived epitopes, further experiments will also be needed to determine whether these results extend to all Mtb T7SS substrates or all Mtb MHC-I antigens, or only a subset.”

Other points that should be addressed:1. The authors should mention the rationale/significance of infecting the macrophages for 72h (rather than 24-48h) in the first set of immunopeptidomics experiments. Mtb infection >48h has been associated with the downregulation of MHC-II in infected cells.

We agree with the reviewer that the time point selected is an important methodological consideration. Based on prior experience in our lab, we knew that infection of hMDMs with Mtb at MOI 2-3 for >72 hours would result in a high rate of cell death, whereas cell viability is high at 72 hours. We did not want to select an earlier time point because antigens transiently presented early in infection might not be protective if T cell responses take multiple days to initiate. Also, we show in Figure 3 —figure supplement 1 that infection with H37Rv does not result in down-regulation of surface MHC-I at 72 hours, which led us to conclude that global down-regulation of MHC-I antigen presentation was not a concern when analyzing antigen presentation in *Mtb*-infected hMDMs at that time point.

We’ve added the following on lines 62-66 to clarify our rationale: “We selected the 72 hour time point over shorter time points because it has previously been reported that some Mtb antigens are expressed only transiently, early in infection (Moguche et al., 2017), whereas Mtb-specific T cell responses may take multiple days to initiate, either upon priming (Chackerian et al., 2002) or during a recall response (Gallegos et al., 2008). Past experience suggested that choosing a longer time point would have resulted in a high rate of cell death.”

2. Hypotheses 1 and 2 are presented in the opposite order in the text (lines 133-134) as in Figure 2a.

We apologize for the oversight. Thanks to the reviewer for pointing this out! We’ve changed the order in which the hypotheses are presented in the text. Lines 140-141 now read “(1) be processed by endolysosomal proteases and loaded onto MHC-I in Mtb-containing compartments, or (2) gain access to cytosolic antigen processing pathways via permeabilization of the phagosome membrane (Grotzke et al., 2010, 2009; van der Wel et al., 2007).”

3. Citation for the statement in line 292

The prior research this statement refers to is described in more detail in the next few sentences and cited there, but we’ve added the appropriate citations to this line (now line 337) as well for the sake of clarity. Thanks to the reviewer for pointing out the ambiguity.

4. It would be helpful if the rationale for (and the difference between) the Esx5 peptides QTVEDEARRMW vs. QTVEDEARRM(ox)W were mentioned in the Results section in addition to the legend for Figure 3. I had to search to understand why only 2 peptides were mentioned in the text, but data for 3 peptides were shown in the figure.

We appreciate the reviewer pointing out this lack of clarity. In lines 205-207 of the text, we added a sentence explaining the rationale for targeting both the oxidized and non-oxidized form of the peptide: “Because methionine residues of peptides can oxidize during sample handling, we targeted both the oxidized and non-oxidized form of EsxJKPW_24-34_ where possible.”

5. MG-132 increased the expression of extracellular and endosomal peptides, but reduced Mtb and nuclear/cytosolic proteins, which is internally consistent and strengthens the authors' conclusions that the inhibitor worked, but did not affect the expression of the two Mtb peptides quantified.

We thank the reviewer for this observation. We agree that the effect of MG-132 on peptides derived from cytosolic and nuclear proteins supports our conclusion that the inhibitor worked, even though we did not see the expected effect on *Mtb* peptides. This was the basis for our conclusion that processing of the *Mtb* peptides quantified in this experiment does not require the proteasome.

6. It is unclear why the authors think that Rv1211 and Rv3196A are candidates for proteins that facilitate protein transport across the phagosomal membrane. This seems to be speculation and completely unsupported. (lines 312)

We apologize for the lack of clarity in this passage and thank the reviewer for pointing out that our wording here was ambiguous. What we meant was not that Rv1211 and Rv3196A might facilitate the transport of other proteins across the phagosome membrane but rather that the fact that Rv1211 and Rv3196A are small, low molecular weight proteins might make it easier for them to diffuse through pores introduced into the phagosome membrane by *Mtb*. We’ve reworded and attempted to clarify this sentence, which now reads “TB8.4, Rv1211, and Rv3196A are all low molecular weight proteins, which we speculate could make it easier for them to translocate through pores in a permeabilized phagosomal membrane.”

7. There are a few typos in the figure legends (sometimes panels are bolded, and sometimes they are not), and sometimes the order of the figure doesn't align with the figure legends.

We thank the reviewer for pointing out these issues. We’ve done our best to proofread the figure legends again, making sure the letters identifying figure panels are always capitalized and bolded, that typos are corrected, and that the letter indices of figure panels match the caption.

We made the following changes to the figure legends:

– Capitalized and bolded “(C)" in Figure 3 caption (line 236)

– Bolded D and E in Figure 4 caption (line 304)

– Capitalized “Figure 1A” in Figure 1 —figure supplement 1 caption

– Changed “supplementary figure 2” to “Figure 1 —figure supplement 2” in Figure 1 —figure supplement 3 caption (line 1248)

– Changed “histograms” to “contour plots” in Figure 3 —figure supplement 1A caption

– Corrected figure panel labels of Figure 4 —figure supplement 1 B-F caption

– Changed “flow cytometry plots” to “flow cytometry contour plots” in Figure 4 —figure supplement 3A caption

8. The manuscript is lacking many of the key citations required to put this work into context. To strengthen this study, the authors should review, cite and discuss their work in the context of the following citations, to provide a clear understanding of what this study adds, and how this study moves the field forward.

We are grateful to the reviewer for taking the time to point us to so many of these references! More specific comments are in-line below.

Some of the references for the Type VII systems need to be altered. None of the references are the first report of the secretion systems. For ESX-1 please cite PMID: 14557536, for Esx-5 please cite PMID: 17076665, and for ESX-3 please cite PMID: 19846780. (lines 92-96)

These references have been added.

The references for cytoplasmic access need revision. Lines 140-121; Please cite PMID: 17604718, for cytoplasmic access; PMID: 22319448, as well as Watson 2012 for ESX-1 dependency.

These references have been added.

The authors need to consider the following references regarding ESX-1 and MHC presentation: PMID: 22022257, which suggests that access to the cytoplasm is required for MHC type 1 presentation in M. tuberculosis; PMID: 18591224, which shows that ESX-1 function is required for priming CD8 T cells, and presenting EsxB peptides, also found here. PMID: 19360129, defining the pathway used by MHC during M. tb infection for phagosomal antigens, PMID: 22901810, showed ESX-1 dependent localization of p62; It is already known that MTb localizes with markers of phagosomal membrane damage in an ESX-1 dependent manner.

We appreciate the reviewer’s guidance on contextualizing the relationship between ESX-1 function and MHC-I antigen presentation. We’ve added a discussion of how the bioinformatic studies in PMID: 22022257 support the functional relevance of presentation of T7SS substrates on MHC-I on lines 426-430, which read “Bioinformatic studies have also shown that predicted class I HLA-binding motifs are statistically underrepresented in several Esx-family protein sequences, suggesting that these proteins may be under selective pressure to escape recognition by CD8^+^ T cells (Maman et al., 2011). This finding provides further support for the idea that T cell recognition of T7SS substrates presented on MHC-I is functionally relevant to protective immunity against Mtb infection.”

We’ve added some additional detail and nuance to our discussion of PMID: 18591224 in lines 340-346 to acknowledge that the authors of this paper argue that ESX-1 is dispensable for non-ESX-1 substrates, but required for ESX-1 substrates. These lines now read “While ESX-1 activity is required for priming of CD8^+^ T cell responses specific for EsxB in vivo in mice, it is dispensable for priming of CD8^+^ T cell responses specific for antigens exported via other secretion systems (such as TB8.4 and EsxH) (Woodworth et al., 2008a). While these prior results suggest that the requirements for presentation of Mtb antigens on MHC-I may vary among Mtb antigens and/or vary among antigen presenting cell types, our results show that ESX-1 activity is essential for the presentation of certain Mtb antigens in infected human macrophages, beyond ESX-1 substrates alone.”

We’ve also made an effort to highlight prior evidence that cytosolic antigen processing is required for presentation of *Mtb* antigens on MHC-I in human dendritic cells. Lines 363-366 now read “Prior studies have shown that Mtb antigens must access the cytosol to be presented on MHC-I in human monocyte-derived dendritic cells (Grotzke et al., 2010, 2009; Lewinsohn et al., 2006), and our results are consistent with a model in which ESX-1-mediated phagosomal membrane damage enables Mtb antigens to access cytosolic antigen processing and presentation pathways.”

We’ve expanded on the passage where we highlight prior results showing ESX-1-dependent phagosome membrane damage to emphasize that the purpose of our microscopy experiments was primarily to demonstrate that this phenomenon was occurring in our primary hMDM model system at timepoints relevant to our MS analyses. Lines 159-161 now read “Our results are consistent with prior studies demonstrating that ESX-1 activity is required for Mtb to damage the phagosome membrane (Augenstreich et al., 2017; Simeone et al., 2012; Watson et al., 2012) and show that this phenomenon occurs in our primary human macrophage infection model.”

The authors should consider these references demonstrating that BCG lacks ESX-1 activity (lines 288-290) References demonstrating BCG is attenuated because it lacks ESX-1: PMID: 10320585 , BCG is missing RD1, PMID: 12410828 RD1 contributes to BCG attenuation, PMID: 12508154 deletion of RD1 from TB leads to attenuation, PMID: 12692540 restoring ESAT-6 secretion to BCG.

Thank you to the reviewers for these additional references. The last reference above (PMID: 12692540) suggested a possible implication of our study that we hadn’t explicitly spelled out in our discussion – namely that. In addition to adding the suggested citations, we’ve added the following on lines 332-335: “The absence of ESX-1 in BCG therefore could limit the ability of BCG to prime effective T cell responses against Mtb, not only because of the absence of antigens encoded in the ESX-1 locus but also because of altered or reduced presentation of other antigens (Pym et al., 2003).”

The authors should review these references (lines 300-302) first suggesting the model that the ESX-1 function precedes that of the other ESX systems (2-5). PMID: 33653883 first suggested that ESX-1 functions to lyse the phagosome, and that ESX-1 activity is followed by the activity of the other ESX secretion systems (see the final model in this paper). ESX-3 was first shown to function downstream of ESX-1 in this paper (PMID: 30482832).

These are both excellent precedents to cite. We’ve expanded our discussion of the interdependence of T7SSs in the intracellular setting as follows (lines 346-355): “Whereas in axenic culture, secretion of ESX-5 substrates is independent of ESX-1 function (Champion et al., 2006; Shah and Briken, 2016), the fact that presentation of peptides derived from an ESX-5 substrate on MHC-I requires ESX-1 activity supports the hypothesis that the localization of secreted Mtb proteins within a host cell may depend on the activity of multiple T7SSs. This dependence on multiple T7SSs has previously been shown for other ESX-5 substrates such as CpnT (Izquierdo Lafuente et al., 2021), and a functional interdependence between ESX-1 and ESX-3 (which respectively damage the phagosome membrane and prevent host membrane repair) has previously been proposed (Mittal et al., 2018). Our results suggest that the interdependence of Mtb T7SSs in an intracellular context influences the availability of Mtb antigens for processing and presentation on MHC-I.”

The original papers reporting these proteins are secreted/ localized should be cited. PMID: 19682254 EspC was first reported as an ESX substrate here. For the list of proteins in 318-320. Please address the rest.

We’ve added a citation to PMID: 19682254. We were unable to find any earlier reports of PPE51, PPE60, or PPE20 being localized to the outer membrane than the references already cited and are convinced these are the earliest reports conclusively showing outer membrane localization of these particular PPE proteins. On line 406 we’ve added citations to PMID: 11800581 and PMID: 17076665, which to the best of our knowledge established (respectively) that PE/PPE proteins can be localized to the outer membrane and that this process depends on type VII secretion systems.

In lines 308-315, did the authors check if other papers that have defined the ESX-1 secretome have seen these proteins? Check PMID: 35671426 for the M. marinum orthologs.

We thank the reviewer for this helpful suggestion! *M. marinum* does not encode a known ortholog of Rv1211, but does encode an ortholog of Rv3196A (MMAR_1367), which is detected in the supernatants of *M. marinum* in PMID: 35671426 in a non-ESX-1-dependent manner. This is consistent with the idea that Rv3196A could be secreted, albeit not by ESX-1. Accordingly, we’ve added a citation to PMID: 35671426 in lines 392-395, which now read “Rv1211 and Rv3196A have both previously been detected by MS in culture filtrates of Mtb (Bell et al., 2012), as has the Mycobacterium marinum ortholog of Rv3196A (MMAR_1367) (Cronin et al., 2022), suggesting that either or both of these proteins could be secreted despite lacking readily identifiable secretion signals.”